# Benchmarking Quantum Reinforcement Learning

**Nico Meyer** [* 1 2]   **Christian Ufrecht** [* 1]   **George Yammine** [1]   **Georgios Kontes** [1]   **Christopher Mutschler** [1]
**Daniel D. Scherer** [1]

## Abstract

Benchmarking and establishing proper statistical validation metrics for reinforcement learning (RL) remain ongoing challenges, where no consensus has been established yet. The emergence of quantum computing and its potential applications in quantum reinforcement learning (QRL) further complicate benchmarking efforts. To enable valid performance comparisons and to streamline current research in this area, we propose a novel benchmarking methodology, which is based on a statistical estimator for sample complexity and a definition of statistical outperformance. Furthermore, considering QRL, our methodology casts doubt on some previous claims regarding its superiority. We conducted experiments on a novel benchmarking environment with flexible levels of complexity. While we still identify possible advantages, our findings are more nuanced overall. We discuss the potential limitations of these results and explore their implications for empirical research on quantum advantage in QRL.

## 1. Introduction

Reinforcement learning (RL) is a powerful algorithmic primitive increasingly applied across multiple domains (Arulkumaran et al., 2017; François-Lavet et al., 2018). Quantum reinforcement learning (QRL) (Meyer et al., 2022) is a collection of RL algorithms developed for quantum computers, an emergent paradigm of computing that exploits the laws of quantum mechanics (Nielsen & Chuang, 2010). While some QRL algorithms with provable advantage over traditional (classical) algorithms have been proposed (Wang et al., 2021; Cherrat et al., 2023; Dunjko et al., 2016), these

---
[*]Equal contribution   [1]Fraunhofer IIS, Fraunhofer Institute for Integrated Circuits IIS, Nürnberg, Germany [2]Pattern Recognition Lab, Friedrich-Alexander-Universität Erlangen-Nürnberg, Erlangen, Germany. Correspondence to: Nico Meyer <nico.meyer@iis.fraunhofer.de>.

*Proceedings of the 42nd International Conference on Machine Learning*, Vancouver, Canada. PMLR 267, 2025. Copyright 2025 by the author(s).

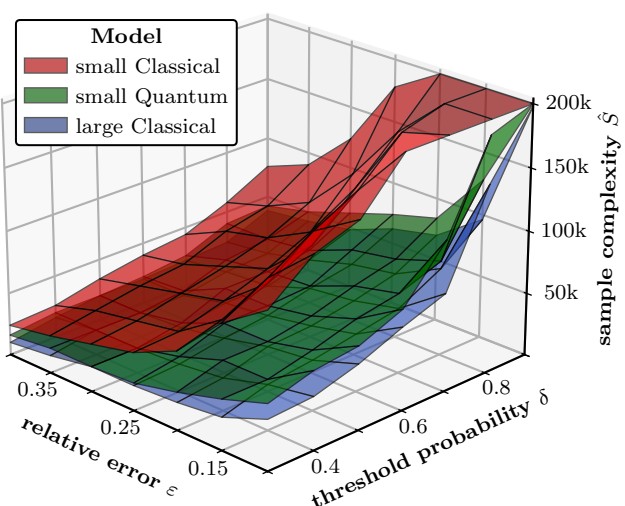

*Figure 1.* Comparison of empirical sample complexities $\hat{S}$ of double deep Q-learning and a quantum version of the algorithm (lower is better). Sample complexity is the number of environment-agent interactions to surpass a performance threshold $1 - \varepsilon$ with probability $\delta$. The figure shows the result for the `BeamManagement6G` environment introduced in this work. In order of decreasing sample complexity: a *small classical* neural network with 2 hidden layers of width 16, i.e., 387 parameters; a *small quantum* circuit with 4 layers on 14 qubits, i.e., 437 variational parameters, integrated between fully connected classical layers with additional 101 parameters; a *large classical* neural network with 2 hidden layers of width 64, i.e., 4611 parameters; The hybrid quantum model consistently outperforms the similar-sized classical network, and is also competitive with the 10-fold larger classical model.

algorithms are currently far out of reach for any existing quantum computing hardware. Hence, most of the work has focused on hybrid algorithms in which the deep neural network of a traditional RL algorithm – either approximating the policy, the value function, or both – is replaced by a variational quantum circuit (VQC) (Bharti et al., 2022; Chen et al., 2020). However, they (while being less hardware-intensive) are heuristic in nature. Hence, there is no proof of intrinsic advantage of the quantum version of the algorithm.

In the search for *heuristic quantum advantage* in QRL, we encounter issues similar to classical RL: sensitivity to hyper-

parameter choices, various randomness sources, and even random seed and codebase (Henderson et al., 2018; Jordan et al., 2024). QRL faces similar, if not more severe, reproducibility issues than traditional RL (Bowles et al., 2024; Franz et al., 2023) due to additional randomness. Therefore, any claim of QRL's superiority over classical RL should be taken with great care.

*"How do we meaningfully assess if a QRL agent outperforms its classical counterpart and what does outperform mean in the context of quantum advantage?"* We adopt *sample complexity*, i.e., the number of interactions between the agent and the environment to achieve a certain performance (Kearns & Singh, 1999; Kakade, 2003), as the central benchmarking metric due to its inherent costly implications in real-world applications. As classical RL is notorious for its sample inefficiency (François-Lavet et al., 2018), a potential quantum advantage in sample complexity presents an intriguing prospect of QRL.

In addition to common sources of randomness known in RL (Henderson et al., 2018), such as random weight initialization, randomness in the environment, etc., QRL is subject to additional sources of randomness such as shot noise or hardware imperfections due to current limitations. Therefore, performance comparisons need to be based on a sound statistical evaluation. However, most studies perform only a small, potentially insufficient number of training runs for robust inferences under stochasticity. As a result, statements are potentially misleading or insignificant. Figure 2 exemplifies the comparison of two RL algorithms w.r.t some threshold on the evaluated return. It illustrates common flaws prevalent in the QRL literature, e.g., averaging learning curves over only 5 seeds, inconsistency in statistical ranges, etc. Figure 2 indicates that algorithm 1 is more sample-efficient. However, the interquartile ranges (shaded areas) estimated with a much larger sample size of 100 runs rather support the opposite statement.

This paper strongly advocates robust stochastic modeling backed by significance testing to meet reproducibility criteria (Henderson et al., 2018). The main contributions of this paper are as follows. (1) We propose a formal evaluation procedure by a statistical estimator for sample complexity, complemented by a robust notion of outperformance based on statistical significance. Importantly, both are designed for the benchmarking of heuristic algorithms. (2) We design and implement a fast and flexible benchmarking suite based on a problem inspired by real-world wireless communication tasks. While community benchmarks (Brockman et al., 2016; Tassa et al., 2018b) are widely used, benchmarking quantum algorithms requires the option to flexibly scale difficulty and instance size of the task to allow extrapolation beyond classical simulatability of quantum algorithms. The task we introduce belongs to a potentially significant prob-

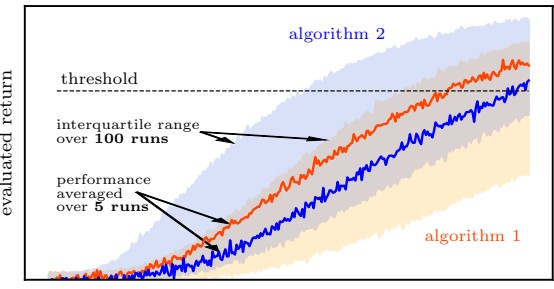

*Figure 2.* Inadequate reporting of two (Q)RL agents' performance can lead to false conclusions about sampling complexity. Although the curves may seem exaggerated, it is common practice in QRL studies to benchmark with such a limited number of runs.

lem class for quantum computing, characterized by small input and output dimensions of classical data (Hoefler et al., 2023). Therefore, we opt for a new benchmark environment to better suit the specific needs of quantum algorithm evaluation. (3) We perform the most extensive computational analysis of classical RL vs. QRL done so far to the authors' knowledge. In our study, we compare different problem and algorithm configurations of different scales and complexity. Statistically robust results are obtained by performing 100 training runs per configuration, by far the largest population size found in the quantum computing literature.

The remainder of this paper is structured as follows. Section 2 reviews related work. Section 3 provides background on (Quantum) RL. Next, Section 4 introduces our statistical estimator. Section 5 describes the experimental setup and introduces a novel environment with flexible complexity. Section 6 evaluates QRL using our statistical estimator. We discuss the implications on quantum advantage in Section 7.

## 2. Related Work

For classical deep RL, in general, no rigorous analysis of algorithms is possible, necessitating a computational approach to algorithmic comparison and representative benchmarking environments (Brockman et al., 2016; Todorov et al., 2012; Tassa et al., 2018a) for mature implementations of widely-used deep RL algorithms (Hill et al., 2018; Duan et al., 2016; Fujimoto et al., 2019; Wang et al., 2019b). However, benchmarking results are heavily influenced by (sometimes seemingly trivial) implementation details (Engstrom et al., 2019; Andrychowicz et al., 2021; Huang et al., 2022), as well as randomness in environment transitions and network initialization (Henderson et al., 2018). Research has focused on determining the necessary number of seeds for robust comparison (Agarwal et al., 2021; Colas et al., 2018) and on adopting statistical analysis over simple point estimates (Agarwal et al., 2021; Colas et al., 2018;

Patterson et al., 2024; Huang et al., 2024) (e.g., reporting the interquartile mean instead of the average performance). However, benchmarking algorithms with increased number of seeds (Laskin et al., 2019; Gorsane et al., 2022; Bettini et al., 2024) can quickly become a bottleneck (Jordan et al., 2024), especially for computationally-intense algorithms, like model-based (Wang et al., 2019b) or safe RL (Zhao et al., 2024). In addition, (Jordan et al., 2020) emphasizes that standard evaluations often overlook the difficulty of hyperparameter tuning, which can obscure the true *usability* of an algorithm in real-world applications.

QRL can solve artificial tasks (without practical relevance) with exponentially smaller sample complexity compared to any known classical algorithm. Recently, (Jerbi et al., 2021; Liu et al., 2021) constructed artificial problems which are widely believed to be classically intractable. When the type of the algorithm is fixed to e.g. policy iteration, quantum versions exist with polynomially reduced sample complexity (Wang et al., 2021; Ganguly et al., 2023; Zhong et al., 2024; Wiedemann et al., 2023). However, these algorithms require resources that far exceed the capabilities of current quantum hardware. An alternative line of research (Chen et al., 2020; Skolik et al., 2022; Lockwood & Si, 2020), more aligned with the limitations of current hardware, substitutes the classical neural network used as a function approximator for policy and value function in classical RL algorithms by a variational quantum circuit VQC (Bharti et al., 2022). Here, Jerbi et al. (2021) construct artificial problems based on the same VQC architecture later used by the learner. The superiority found empirically for the QRL algorithm can then be attributed to the inductive bias introduced in the problem. Several studies have compared classical RL and QRL on toy problems inspired by real-world tasks. Some of these studies (Chen et al., 2020; Drăgan et al., 2024; Reers, 2023; Hohenfeld et al., 2024; Eisenmann et al., 2024) have documented empirical superiority of QRL on metrics closely related to sample complexity. This work proposes a robust methodology to compare these types of algorithms.

## 3. (Quantum) Reinforcement Learning

RL is a framework for solving complex time-dependent decision-making problems. It relies on a Markov Decision Process (MDP), represented as a 5-tuple $(\mathcal{S}, \mathcal{A}, R, p, \gamma)$, where $\mathcal{S}$ is the set of states and $\mathcal{A}$ is the set of actions. The reward function $R : \mathcal{S} \times \mathcal{A} \times \mathcal{S} \mapsto \mathbb{R}$ assigns a scalar value to performing action $a$ in state $s$ and transitioning to state $s'$. The dynamics are governed by $p : \mathcal{S} \times \mathcal{S} \times \mathcal{A} \mapsto [0, 1]$, which gives the probability of transitioning from state $s$ to state $s'$ after taking action $a$. The discount factor $\gamma$, ranging between 0 and 1, determines the importance of immediate versus future rewards. The objective is to find a policy $\pi(s) = a$ that maximizes the discounted long-term reward $G_t \leftarrow$

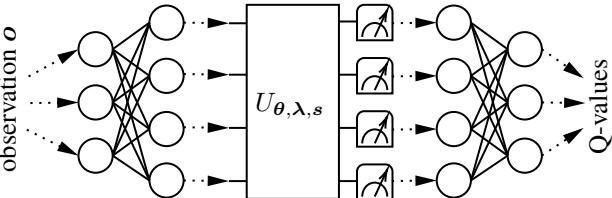

*Figure 3.* Hybrid classical-quantum neural network for an exemplary 4-qubit quantum layer. The dimensionality of the observation is mapped to the number of qubits in the quantum circuit using a fully-connected layer. The variational quantum circuit – for details on the ansatz see Figure 18 in Appendix D – acts as a hidden layer, and all qubits are measured individually in the Pauli-Z basis. These measurement results are post-processed using another fully-connected layer, mapping to the number of actions.

$\sum_{t'=t}^{\infty} \gamma^{t'-t} r_{t'}$ from time step $t$ onwards. With the state-action value function $Q_\pi(s, a) := \mathbb{E}_\pi [G_t | s_t = s, a_t = a]$, the optimal policy is given by $\pi^*(s) = \mathrm{argmax}_a Q^*(s, a)$. The optimal Q-value function is the unique solution to the Bellman optimality equation $Q^*(s, a) = \sum_{s'} p(s'|s, a) [R(s, a, s') + \gamma \cdot \max_{a'} Q^*(s', a')]$, for all $s \in \mathcal{S}$, $a \in \mathcal{A}$ (Sutton & Barto, 2018).

The state-action value function is typically represented using some type of function approximator, as tabular approaches are only suitable for small problem instances. Reinforcement learning based on classical deep neural networks (DNNs) has been first successfully realized in the deep Q-networks (DQN) algorithm (Mnih et al., 2015). In our work, we employ an extension of this concept, i.e. double deep Q-networks (DDQN) (Van Hasselt et al., 2016). This algorithm employs function approximators $Q_\theta, Q_{\theta'}$, and performs updates of the parameters towards following loss:

$$\mathcal{L}(\theta) = [r + \gamma \cdot Q_\theta(s', \mathrm{argmax}_b Q_{\theta'}(s', b)) - Q_\theta(s, a)]^2 ,$$

where target parameters $\theta'$ are synchronized after a hyperparameter-dependent number of update steps.

It is also possible to additionally parameterize the policy (i.e. the actor), in addition to the value function (i.e. the critic), and train the respective function approximators with proximal policy optimization (PPO) (Schulman et al., 2017). Details on both the algorithm and the hyperparameter tuning performed can be found in Appendix C.

We compare two function approximation approaches in these algorithms: (i) the standard version with a classical DNN and (ii) the quantum version, which has a variational quantum circuit between two small classical fully-connected layers. This structure is called *hybrid classical-quantum*, as depicted in Figure 3. As discussed in Appendix D.2, the classical layers have few trainable parameters, and small fully classical DNNs are not competitive, so the hybrid models'

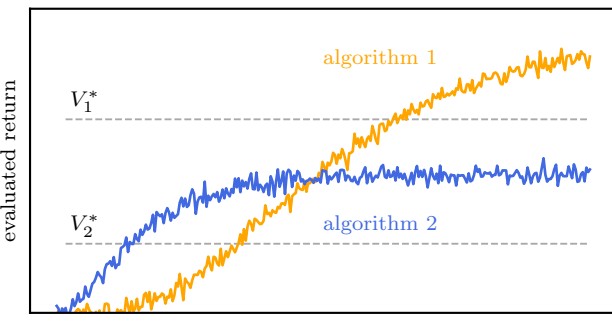

*Figure 4.* The figure exemplarily shows two learning curves generated by two different algorithms or algorithmic settings (algorithm 1 and algorithm 2). While algorithm 2 exhibits lower sample complexity with respect to threshold $V_2^*$ than algorithm 1 (for this particular training run), the converse is true for threshold $V_1^*$, which algorithm 2 may even never reach. Consequently, if convergence to optimality cannot be proved for the algorithm, sample complexity is well defined only with respect to a given threshold.

performance originates from the quantum sub-module. The VQC acts like a quantum counterpart to a classical DNN, with trainable weights parameterizing unitary operations, usually single-qubit rotations (Bharti et al., 2022). The quantum state is measured to approximate complex functions. Past research suggests VQCs may have advantages over DNNs, including better accuracy for certain tasks (Liu et al., 2021) and smaller model size (Chen et al., 2020). More details on VQCs and the ansatz used here are shown in Appendix D.1. Such models have been used in value-based (Chen et al., 2020; Skolik et al., 2022) and policy-based (Jerbi et al., 2021; Meyer et al., 2023a) RL routines.

## 4. Sample Complexity Estimator

The sample complexity $S$ of an RL algorithm is the number of samples $s' \sim p(\cdot|s, a)$ required to meet a specified performance criterion (Kearns & Singh, 1999; Kakade, 2003) with high probability. When $S$ is expressed by relevant problem parameters such as state- and action-space size, discount factor etc., it serves as an effective instrument to compare different learning algorithms with performance guarantees (Lazaric et al., 2012; Lattimore et al., 2013; Liu et al., 2024). However, assessing sample complexity of heuristic RL algorithms requires empirical evaluation procedures. In particular, as illustrated in Figure 4, sample complexity for algorithms without performance guarantees has to be defined with respect to a given threshold $V^*$. In the following, we introduce a statistical estimator for sample complexity. To this end, for a given algorithm, we view each training run as the realization of a stochastic process $\{V_t, t = 1, ..., T\}$. In our setting, we choose $V_t$ to be the

evaluated expected return. In case of our environment, the maximal value of $V_t$ can be calculated, thus without loss of generality, we constrain $V_t \in [0, 1]$. We generalize the definition of the estimator in Appendix B to environments for which the optimal value of the expected return is unknown. We fix $\delta \in (0, 1]$ and $\varepsilon \in [0, 1]$, and define sample complexity as

$$S = \sum_{t=1}^{T} \mathbb{I}\left[P_t < \delta\right], \quad (1)$$

where $\mathbb{I}[\cdot]$ is the indicator function which is 1 exactly if the probability $P_t = P(V_t \geq 1 - \varepsilon)$ is smaller than $\delta$ and otherwise 0. In words, $P_t$ is the probability that the algorithm performs better than the threshold $1 - \varepsilon$ at time step $t$. Since $P_t$ is unknown, we model $N$ training runs by a collection of stochastic processes $\{V_t^{(i)}, t = 1, ..., T\}$ with $i = 1, ..., N$, where $V_t^{(i)}$ i.i.d. for given $t$. Next, we replace $P_t$ by its unbiased estimator

$$\hat{P}_t = \frac{1}{N} \sum_{i=1}^{N} \mathbb{I}\left[V_t^{(i)} \geq 1 - \varepsilon\right]. \quad (2)$$

Now the empirical sample complexity can be defined as:

**Definition 4.1** (Estimator empirical sample complexity). Given $N$ training runs $\{V_t^{(i)}, t = 1, ..., T\}$ with $i = 1, ..., N$ and $V_t^{(i)}$ i.i.d. for given $t$, a probability threshold $\delta \in (0, 1]$ and a performance threshold value $\varepsilon \in [0, 1]$, we call

$$\hat{S} = \sum_{t=1}^{T} \mathbb{I}\left[\hat{P}_t < \delta\right], \quad (3)$$

where $\hat{P}_t$ is defined in Equation (2), the *empirical sample complexity*.

Appendix B provides more details on the intuition of this definition. Additionally, we prove consistency, that is $\lim_{N \to \infty} P(|\hat{S} - S| > \eta) = 0$ for $\eta > 0$. Moreover, using the central-limit theorem, it is shown that $\hat{S}$ is asymptotically unbiased. Based on the analysis in the appendix we choose $N = 100$ throughout this work. To the author's knowledge, this number far exceeds the population size used in any other study on QRL benchmarking. Based on Definition 4.1 we define:

**Definition 4.2** (Significant Outperformance). We say that algorithm 1 outperforms algorithm 2 [on a task], if (i) it has significantly lower sample complexity (with respect to a definition of significance) for some error threshold $\varepsilon$ and probability threshold $\delta$. (ii) Algorithm 1 must not have a significantly higher sample complexity than Algorithm 2 for any $\varepsilon$-$\delta$-configuration.

In this work, we consider a difference in sample complexity to be significant, if the respective 5th and 95th percentile

ranges do not overlap (i.e., extended interdecile ranges (De-Groot, 2005)). To guarantee robustness, we perform 100 runs for each setup and use cluster re-sampling (Cameron et al., 2008) for estimating the quantiles.

## 5. Experimental Setup

As QRL algorithms exhibit some robustness against hardware noise (Skolik et al., 2023) (which is expected to further decrease in the future (Kim et al., 2023; Acharya et al., 2025)) we decided to perform all experiments in a noise-free simulation. We reduced the impact of parameter initialization by considering 100 random seeds per environment and model configuration. We initialized the classical neural networks using He initialization and the VQC parameters uniformly at random within $[0, 2\pi]$. We tuned the hyperparameters, see Appendix C, before running the actual experiments. The configurations of the models are reported in terms of three metrics: (i) model *width*, i.e., the number of neurons in hidden layers for classical DNNs, and in case of quantum architectures the number of qubits; (ii) model *depth*, i.e., the number of hidden layers for classical networks, and the number of layers of the quantum models; and (iii) model complexity, parameterized by the number of trainable parameters, which will be justified further below. The reported configurations were chosen by an extensive ablation study, see Appendix D.

**BeamManagement 6G Environment.** Solutions to 'industrial use cases' with quantum machine learning (QML) (Dunjko & Briegel, 2018) and QRL (Meyer et al., 2022) are currently limited to toy problems, far from generating commercial value. This limitation is due to constraints of current hardware (i.e., the number and quality of qubits) and the input-output bottleneck (Hoefler et al., 2023).

We propose a novel benchmarking environment that focuses on beam management in wireless communication. Next-generation communication networks feature antennas capable of forming directional beams to serve mobile phones (Enescu, 2020). Beam management is the task of selecting the antenna and beam direction that maximize beam quality (its intensity) at the position of a moving phone. The (discretized) antennas and beam directions are *precoded* into a codebook.

Promising solutions to beam management are based on RL (Wang et al., 2019a; Yammine et al., 2023). Without explicit knowledge of the specific trajectory of the mobile phone, the RL agent is trained to select optimal antenna index and codebook element, only given the selections of previous time steps. This describes the RL state space as

$$\mathcal{S} = \text{Antenna} \times \text{Codebook} \times \text{Intensity},$$

where Antenna is the set of antenna indices, and Codebook is the set of codebook elements. As input to the model,

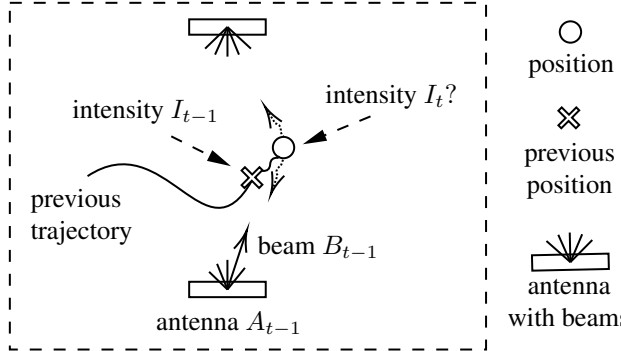

*Figure 5.* The `BeamManagement 6G` environment consists of a set of antennas $A \in$ Antenna, for which at any point in time only one is active. Furthermore, each antenna is equipped with multiple beams, also referred to as *codebook element* $B \in$ Codebook, which are selected automatically. A user moves through the environment, is targeted by one of the antennas, and receives some intensity $I \in$ Intensity. Based on this observation, i.e., the active antenna $A_{t-1}$, beam $B_{t-1}$, and intensity $I_{t-1}$ at the previous time step $t-1$, the task is to select the optimal antenna for the next timestep $t$, i.e., the $A_t$ providing the greatest intensity value $I_t$ to the user. The objective is to maximize the sum of received intensities over the entire trajectory. Note, that the spatial position of the user is unknown, as localization induces unreasonable real-world overhead, and furthermore collides with user privacy concerns.

both antenna and codebook indices are re-scaled, and the intensity value is in $[0, 1]$ by construction in our model. Following this paradigm, we developed a fast simulator that allows flexible placement of multiple antennas and to sample random movement trajectories of varying complexity. For simplicity we task the agent to select the base station (i.e. the antenna) but assume the optimal codebook element to be found automatically by the antenna, i.e.,

$$\mathcal{A} = \text{Antenna}.$$

Selecting a (close to) optimal beam can be solved via efficient beam search algorithms, cf. (Yammine et al., 2023). The reward is the intensity received after selecting the respective antenna and codebook element. A sketch of this environment is given in Figure 5, details are deferred to Appendix A and Appendix E.

We specifically employed this environment because it has a small state and action space, yet exhibits complex dynamics, enabling meaningful analysis without overwhelming encoding complexity. Moreover, its state space is mostly continuous, reflecting realistic energy variations and beam selections as continuous angles. Finally, there is ongoing debate on suitable methods for beam management, with RL identified as a strong contender (Maggi et al., 2024; Voigt et al., 2025), making this environment an excellent choice for benchmarking classical RL and QRL in a set-

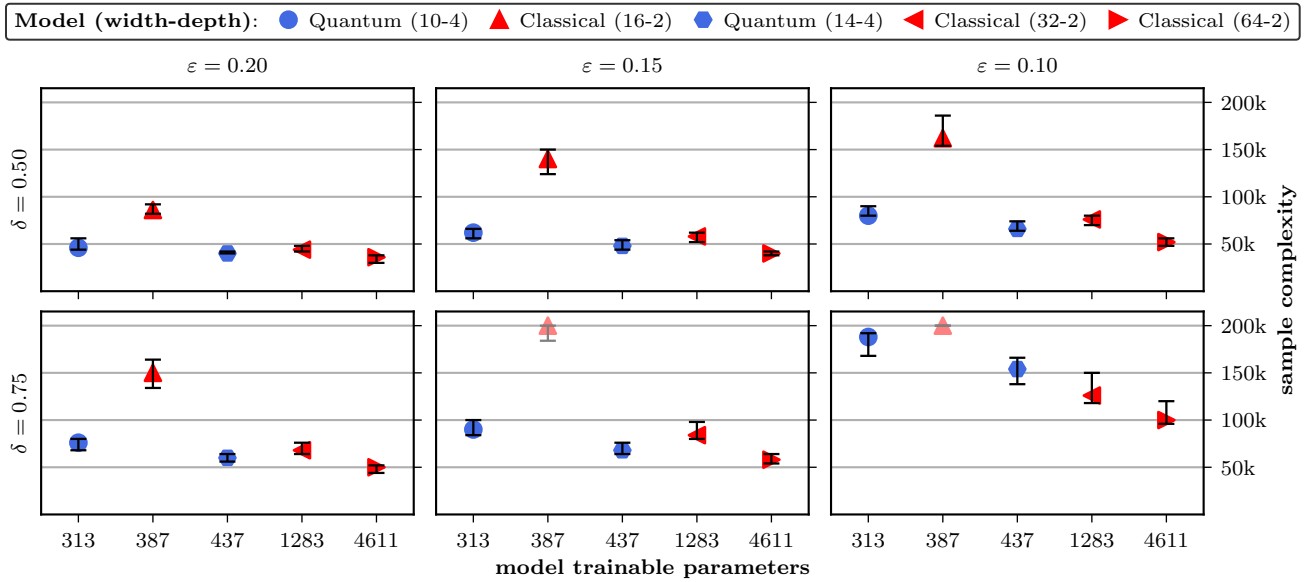

*Figure 6.* Empirical sample complexities $\hat{S}$ of double deep Q-learning for various relative errors $\varepsilon$ and threshold probabilities $\delta$ on the `BeamManagement6G` environment. The model width denotes the number of neurons in classical hidden layers, and the number of qubits in quantum models, respectively. Moreover, model depth refers to the number of hidden layers in DNNs, and number of ansatz repetitions in VQCs. The estimate is based on 100 training runs, with a validation granularity of 2000 steps. The error bars denote the 5th and 95th percentiles, estimated with cluster resampling. The results follow the performance pattern already highlighted in Figure 1. Both quantum models are on par with the 3- to 4-fold larger medium-sized classical model, with the 14-qubit model having a slight edge. Configurations where multiple runs do not achieve the targeted relative error rate before the cut-off 200k steps are shown transparent.

ting inspired by practical 6G challenges. Nonetheless, we highlight that this environment is merely one example to empirically demonstrate our methodology, which can also be applied to other, entirely different tasks.

## 6. Experiments

We evaluate different (Q)RL algorithms on our novel `BeamManagement6G` environment using our proposed statistical sample complexity estimator in Section 6.1. Section 6.2 studies the relationship between sample complexity and (RL) model complexity. In Section 6.3 we compare the performance of the trained policies.

### 6.1. Sample Complexity

We extract the sample complexity of a model via post-processing of the validation logs for 100 randomly seeded runs. A training epoch collects trajectories from 10 environments, each with a horizon of 200 steps. Training is conducted for 100 epochs in total, i.e., overall $200,000$ agent-environment interactions are conducted in each run. After each epoch, validation is performed on 100 environment instances, reporting the ratio of received beam intensity vs. the optimal intensity. This validation with relative intensities is done to enhance the stability of the estimate, and allow for

evaluating the sample complexity depending on the relative error threshold $\varepsilon$. In contrast, the agent itself has access only to the intensity values, in accordance with the real-world conditions. For practical reasons, we evaluate the sample complexities on only the subset of $\varepsilon$-$\delta$-configurations, which are meaningful for real-world performance.

In Figure 1, we plot the empirical sample complexity for three instances of the DDQN algorithm on the standard configuration of three antennas of the `BeamManagement6G` environment. Overall, the large classical model with $4611$ trainable parameters (width 64, depth 2) exhibits the lowest, i.e., best, sample complexity. The small quantum model with only 336 classical and quantum parameters (14 qubits, 4 layers) nearly matches this performance. In contrast to that, the approximately equally-sized small classical model with 387 parameters (width 16, depth 2) clearly requires much more samples for convergence. This hierarchy is preserved across a wide range of threshold probabilities $\delta$ and error thresholds $\varepsilon$. Note, that a sample complexity of $200,000$ indicates, that some runs failed to converge to the desired error threshold.

We extend this analysis by conducting cluster resampling (Cameron et al., 2008), a variant of bootstrap resampling which captures the correlations in time of the learning process. We employ this method to estimate the 5th and

95th percentiles of the estimator $\hat{S}$ for sample complexity. The percentile ranges are indicated by error bars throughout this work, and used to determine outperformance according to Definition 4.2.

In Figure 6 we summarize our main results, and additionally consider two more models: An even smaller quantum model with 313 parameters (10 qubits, 4 layers). It exhibits a performance close to, but not quite competitive with the 14-qubit hybrid model. The restriction to 10 qubits in most experiments optimally utilizes limited computational resources, see Appendix D.2. Overall, we conclude, that hybrid classical-quantum models significantly outperform similar-sized fully classical approaches w.r.t. sample complexity. Furthermore, we include a medium-sized classical model with $1,283$ parameters (width 32, depth 2), which performs slightly worse than the large quantum model for most, but not all $\varepsilon$-$\delta$-configurations. Moreover, the large quantum model is not significantly outperformed by the largest DNN, demonstrating signs of competitiveness.

A similar analysis of the PPO algorithm, where both policy and value function are approximated with DNNs and VQCs, respectively, supports our findings. Details can be found in Appendix C.2. While the results qualitatively match previous observations, typical sample complexities are much higher across all models. However, this is not surprising, as on-policy approaches like PPO are known to be less sample-efficient than off-policy routines like DDQN. This superiority stems from structures like, e.g., the experience replay buffer, which allows the latter algorithm to re-use previous experience (Sutton & Barto, 2018).

### 6.2. Scaling with Model Complexity

We now explore the relationship between sample efficiency and the complexity of the RL and QRL model, respectively. There exist different measures for model complexity both in the classical (Hu et al., 2021) and quantum (Abbas et al., 2021) domain. In this work we define model complexity based on the number of trainable parameters. The parameter count defines a sequence of models in model space (assuming additional hyperparamter search for given parameter count). This sequence can then be used to identify potential trends, for example in the behavior of sample complexity as we scale to more powerful quantum models.

In the previous section, we experimentally demonstrated that a hybrid classical-quantum model significantly outperforms purely classical models of similar complexity and competes with much larger ones. In Figure 7, we analyze this behavior more systematically by plotting sample complexity against both classical and quantum model complexities. Specifically, we vary the width of hidden layers for the classical DNNs, and the number of qubits for VQCs. A deeper analysis including investigations into different model depth can

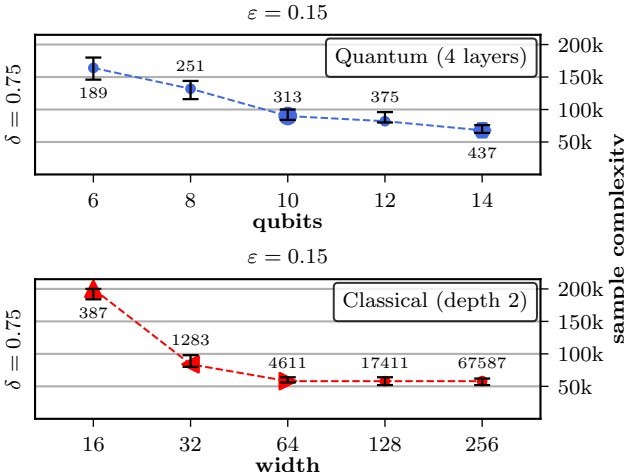

*Figure 7.* Correlation of model and sample complexity in the `BeamManagement6G` environment with the DDQN algorithm. The upper plot depicts quantum models with increasing number of qubits in the variational quantum circuit, the lower plot shows classical DNNs with increasing width of the hidden layers. The number next to the markers denotes the number of trainable parameters in the respective model. All results are averaged over 100 seeds, and error bars denote the 5th and 95th percentiles, estimated with cluster resampling. A more extensive version of this plot can be found in the appendix in Figure 19.

be found in Appendix D.2.

Most crucially, the sample complexity of the classical models saturates once the hidden layer width reaches 64. In contrast, increasing the qubit number of the quantum model up to 14 does not exhibit a similar saturation behavior.

At this point, two questions arise which are addressed in the following: (i) Are models with lower complexity but similar performance generally preferable, i.e., how should sample and model complexity be balanced? (ii) Is outperfomance of the quantum model achievable by further increasing qubit numbers? The first question (i) cannot be answered in general as it depends on concrete practical considerations and objectives. If the primary goal is to minimize the number of agent-environment interactions, regardless of training and inference costs, the classical models are superior. However, when memory requirements are considered, the reduced parameter count of the quantum-enhanced model might be advantageous. Assuming scalability of this approach, also runtime improvements are conceivable. The second question (ii) cannot be answered conclusively at the current stage. The saturation behavior in Figure 7 suggests that further scaling the quantum model could lead to comparable or superior performance relative to the largest classical model, which, however, can only be substantiated by more experimental evaluations. We stress that a naive extrapola-

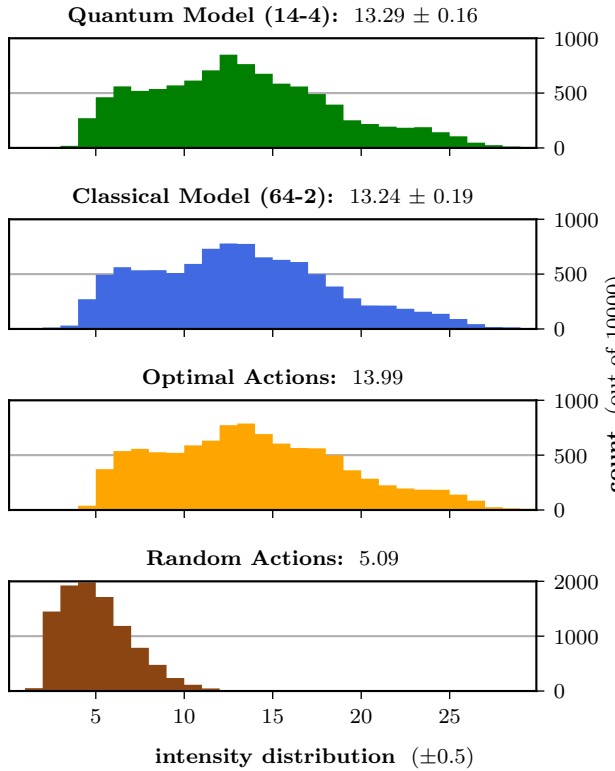

*Figure 8.* Cumulative intensities over trajectories achieved in the `BeamManagement6G` environment. For both, the trained hybrid quantum and classical model, we select the 10 best-performing instances out of the trained models and evaluate each on 1000 random trajectories. The quantum model (green) and the classical model (blue) closely match the ground-truth performance (yellow), with a large performance gap to random behavior (brown). Next to the subplot identifiers we report the mean value of observed intensities over the 10 model instances.

tion to higher qubit numbers might be limited in practice by trainability issues (Larocca et al., 2024). Moreover, it is computationally prohibitive to generate simulation results beyond 14 qubits due to the substantial overhead required for statistically robust results as summarized in Table 3 in Appendix D.2.

### 6.3. Performance of Trained Policies

In the following, we validate that the RL models trained in Section 6.1 behave in a meaningful way. Therefore, we identified the 10 best-performing instances with width 64 and 14 qubits, respectively. In Figure 8, we report the histogram of intensities for the standard environment configuration (see Figure 11) received by employing the respective policies on 1000 random trajectories each.

As a consequence of different trajectories, the maximum achievable intensity varies. Both RL strategies produce

similar-looking histograms of the received intensities. The reported mean values of the intensity of the quantum model (green) and the classical model (blue) do not indicate significantly different behavior by visual inspection. Furthermore, we observed comparable results for various different environment configurations in Appendix E. Therefore, we conclude that the performance of the QRL and classical RL models is on par for the `BeamManagement6G` environment.

These closely match the ground-truth intensity distribution (yellow) obtained by brute force. Moreover, all approaches clearly improve upon a random strategy. Note that the policies of the trained models correspond to non-trivial behavior patterns. We stress that simply selecting the closest antenna is sub-optimal, as highlighted by the complex spatial intensity patterns shown in Figure 13 in the appendix. Moreover, as highlighted in Figure 5, the position of antenna and the mobile phone are unknown to the agent and only learned implicitly.

### 6.4. Standard CartPole Benchmark

To further test our benchmarking scheme, we extend our analysis to the widely used CartPole-v1 environment. This benchmark regularly appears in QRL studies (Lockwood & Si, 2020; Skolik et al., 2022), often accompanied by claims that quantum models outperform classical approaches with fewer number of trainable parameters. However, these statements rely on a small number of training runs and are based on simple visual comparisons. In contrast, here we apply our statistical sample complexity estimator to CartPole-v1 with the results shown in Figure 9.

We employed a vanilla policy gradient algorithm and performed hyperparameter search over a similar range as reported in Appendix C.1. While generally we do not expect to find globally optimal settings (Jordan et al., 2020), more advanced training methods such as natural gradients (Meyer et al., 2023b) could potentially further improve performance. Nevertheless, the chosen configurations allows us to compare classical and quantum function approximators with the number of trainable parameters ranging from about 30 to 17,000. The best quantum setup found, a single-layer model on only four qubits (Meyer et al., 2023a), performs competitively to the best classical model across nearly all $\epsilon$-$\delta$ configurations. The cross-cut, Figure 9(b), of Figure 9(a) at an evaluated threshold of $\varepsilon = 0.05$ shows that the performance gap may not always be statistically significant due to overlapping confidence bounds. This again underscores the importance of sufficient number of runs.

We stress that these results should not be interpreted as evidence for quantum advantage. The circuit sizes are very small, allowing straightforward classical simulation, and scaling CartPole-v1 in a way that requires larger quan-

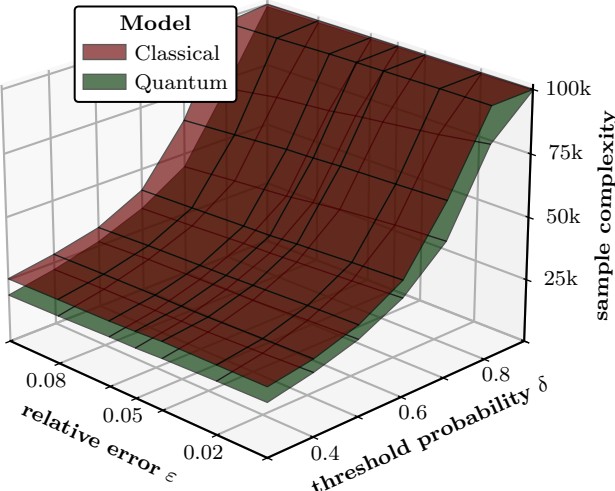

(a) Surface plot of sample complexity over various $\epsilon$-$\delta$ configurations. The classical model contains one hidden layer of width 16, the quantum model operates on 4 qubits.

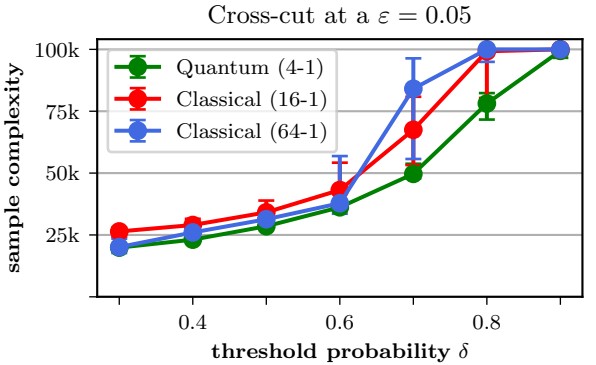

(b) A cross-cut of sample complexity at a relative error of $\varepsilon = 0.05$. The error bars denote the 5th and 95th percentiles.

*Figure 9.* Comparing empirical sample complexities $\hat{S}$ of vanilla policy gradients and a quantum version of the algorithm on the `CartPole-v1` environment. Both classical and quantum models were optimized over a wide range of hyperparameters and model sizes. The estimates are based on 100 training runs.

tum models is notoriously difficult. These limitations motivate our focus on `BeamManagement6G`, whose complexity can be flexibly scaled.

## 7. Discussion of Quantum Advantage

In this section, we discuss the relation of our benchmarking results of the previous section to potential quantum advantage. Let us define *heuristic quantum advantage* as the consistent and statistically significant (Definition 4.1) outperformance (Definition 4.2) of a quantum algorithm over the best (known) classical algorithm for a given task. *Do*

*our findings show signatures of heuristic quantum advantage?* Not quite, because despite extensive hyperparameter optimization, we cannot guarantee that we compared the quantum model to the optimal classical one.

If we set aside this caveat for the moment as double Q-learning is known to be comparatively sample-efficient (Sutton & Barto, 2018), the results are more nuanced. If we steadily increase the number of parameters of the classical model, our numerical results demonstrate that its performance approaches that of the quantum model, but plateaus before significantly outperforming it. Thus, *did we verify quantum advantage under the condition of comparable parameter numbers between models?* Here, we caution again because a necessary condition for any kind of quantum advantage is classical intractability.

Evidence for quantum advantage obtained in small-scale experiments must therefore necessarily be accompanied by arguments for its persistence as problem sizes increase. For example, from our studies on model complexity we might extrapolate that the performance of the quantum model could be further improved by increasing the number of qubits. In this sense, small-scale experiments may indicate trends that might or might not persist for large problem instances. We therefore encourage more experiments and simulations with the striving to scale, which might offer valuable insights into the behavior of quantum algorithms. Ultimately, the question of empirical quantum advantage is, at its core, an empirical one and an answer will come from experimental progress of the future.

## 8. Conclusion

In summary, we introduced a robust and statistically sound methodology for benchmarking heuristic RL, in particular quantum RL algorithms. Our procedure relies on an empirical notion of sample complexity captured by a statistical estimator and a rigorous definition of outperformance. We developed a benchmarking suite inspired by real-world wireless 6G communication tasks, which is flexibly adjustable in difficulty and instance size.

As an application of our methodology, we compared the performance of double deep Q learning, as well as proximal policy optimization, and their quantum counterparts. In an extensive and statistically robust computational analysis, covering many structurally different problem instances and models, we found that the quantum algorithm consistently outperforms the classical version when the number of trainable parameters is similar.

We evaluated the results with respect to potential empirical quantum advantage. We argued that currently no definitive statement can be made but identified trends, that may be corroborated by further experiments on larger scale.

## Acknowledgements

We thank D. Fehrle for helpful discussions. The authors gratefully acknowledge the scientific support and HPC resources provided by the Erlangen National High Performance Computing Center (NHR@FAU) of the Friedrich-Alexander-Universität Erlangen-Nürnberg (FAU). The hardware is funded by the German Research Foundation (DFG). Composing and producing the results presented in this paper required about 40000 core hours of compute.

**Funding:** This research was conducted within the Bench-QC project, a lighthouse project of the Munich Quantum Valley initiative, which is supported by the Bavarian state with funds from the Hightech Agenda Bayern Plus.

## Impact Statement

This paper aims to advance the field of Quantum Reinforcement Learning by introducing a statistical benchmarking routine for rigorous evaluation. Our work addresses the need for substantiated claims of quantum advantage, promoting a standard of thorough assessment. We anticipate that this contribution will enhance the quality and reliability of future research in Quantum Reinforcement Learning, supporting the development of robust and verifiable advancements in the field.

## Data Availability

Implementations of environments, algorithms, and evaluation routines described in this paper are available in the repository https://github.com/nicomeyer96/qrl-benchmark. The framework allows for full reproducibility of the experimental results in this paper by executing a single script. Usage instructions and additional details can be found in the README file. Further information and data is available upon reasonable request.

## Statement of Independent Work

We acknowledge the existence of a research paper with the same title as ours (Kruse et al., 2025). We wish to clarify that both works were conducted independently and without knowledge of each other.

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

# A. 6G Beam forming and RL environment

This appendix elaborates on the 6G beam management environment used in this work as a benchmark environment for different quantum and classical reinforcement learning algorithms.

Beamforming refers to the technique of transmitting signals in a spatially selective manner despite interference and noise. There are multiple practical applications for radar and sonar, in communication technology, radio astronomy, seismology and many more (Trees, 2002). In recent years, the interest in beamforming has further increased due to applications in wireless communication. Beamforming is seen as one of the key technologies to accommodate the growing number of users of high data rate services (Corici et al., 2021). Next generation communication networks will feature antennas (in the following also referred to as base stations) capable of forming directional beams and spatially targeting user equipment (UE). This process is known as *beam management*.

In this work, we consider one specific optimization task, known as hand-over management, the task of switching base stations to maintain optimal quality of service for mobile UEs. For given positions of base stations, the transmitted spatial radiation intensity pattern is typically highly complex due to reflection and absorption effects. Selecting the optimal base station (the one with the highest beam intensity at the UE's location) would require knowledge about the exact spatial intensity pattern and geolocation of the UE based on initial high-resolution measurements of the intensity field. Machine learning, particularly reinforcement learning (Gershman et al., 2010), offers a promising simplification by implicitly learning the intensity field and UE position. Different antenna settings (such as transmission angle, phases and amplitudes of the transmitters,...) are encapsulated in a codebook–a discretized mapping between these intrinsic antenna settings and resulting macroscopic intensity patterns. In general, the RL agent's task is to select the antenna and codebook element which achieves optimal service.

The practical problem sizes of interest far exceed what can be addressed by currently available quantum computing technology. Thus, we employ a simplified toy model with the underlying physics simulator taken to be as realistic as possible. This was ensured by cross-validating our simplified model with a general physics simulator (Burkhardt et al., 2014) for this problem. Our setting is confined to two spatial dimensions, ignoring reflection, absorption, and beam interference. We assume one beam per antenna but allow a variable number of base stations. The task is to train a reinforcement-learning agent that selects the base station with the greatest radiation intensity at the current location of the agent at each time step. We assume that the optimal beam direction (that is the codebook entry) of the selected base station is automatically determined by the antenna.

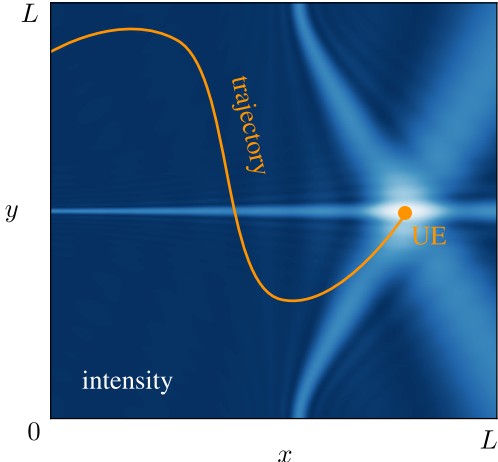

*Figure 10.* A typical setting in the toy-model reinforcement-learning problem inspired by 6G beam management tasks. The UE moves on a randomly generated trajectory (orange line) in the intensity field of three base stations at positions $(0, L/2), (L/2, 0), (L/2, L)$. The main lobe of the antennas is focused on the UE (darker blue corresponds to lower intensity). Clearly visible are also the weaker maxima of the intensity fields stemming from interference effects within the antenna. By traversing the area between the base stations multiple times on different trajectories the reinforcement learning agent learns to select the base station with highest intensity without knowledge of the spatial intensity distribution and the form of its trajectory. For better readability, in this figure the typical inversely squared decrease of the electric field with the distance to the source is removed.

The agent only has access to the following information:

- the index of the base station $A_{t-1} \in$ Antenna selected in the previous time step $t-1$, where Antenna is the set of antenna indices,

- the active codebook element $B_{t-1} \in$ Codebook of the base station selected in the previous time step, where Codebook is the set of codebook elements per antenna, and

- the received radiation intensity $I_{t-1} \in$ Intensity from the previously selected base station.

Note that the agent neither has access to its current position in the radiation field nor to the specific structure of the beam intensity field. While moving along a randomly generated trajectory, the agent learns the mapping between this available information and the spatial beam intensity pattern and its current location. Figure 10 visualizes a typical intensity field in our toy model, showing the trajectory of a UE and the beams from three base stations directed towards the UE. Figure 11 exemplarily details an environment instance with three antennas and shows the ground-truth for the selection task.

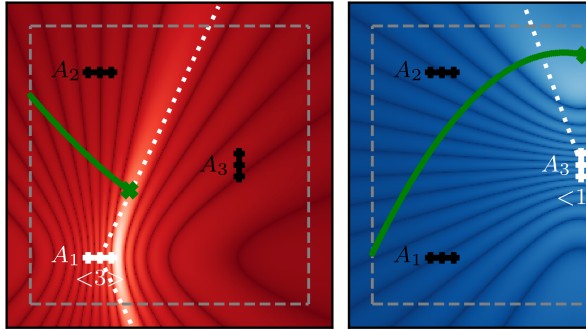 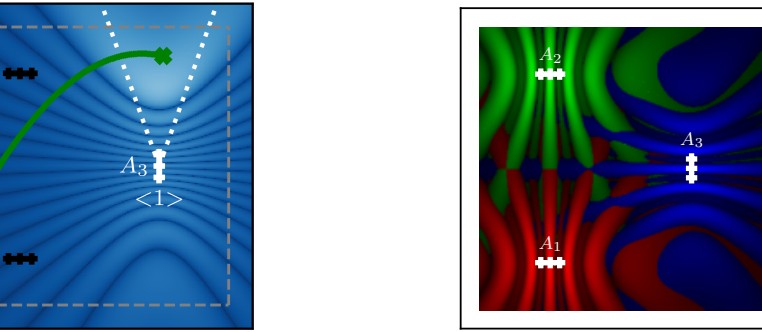

(a) Two instances of a three-antenna environment configuration, with different user trajectories. The dashed gray lines depict the environment boundaries, while the white and black markers indicate the position and orientation of antennas $A_1$, $A_2$, and $A_3$. The trajectories are indicated by the green lines, the green crosses mark the current positions of the user. The task is to select the antenna that leads to optimal service (greatest beam intensity) at the position of the user. The active antenna is marked in white, with the active codebook element indicated by $\langle \cdot \rangle$. The (symmetric) discretized directions of the main beam are indicated by the white dotted lines. The secondary beams are caused by interference effects, but can also be used to serve users, significantly complicating the task of optimal selection. In each timestep, the agent has to base its decision solely on the previously selected antenna, codebook element, and observed intensity, without access to the spatial position of the user.

(b) Ground-truth of the optimal intensity distribution, generated by brute-forcing over all available antenna-codebook configurations. At each point, only the highest received intensity at each point is reported. The colors indicate the optimal antenna, the brightness the intensity magnitude. We emphasize the non-triviality of the ground-truth solution, i.e. the optimal antenna selection does not just correspond to selecting the antenna with the smallest spatial distance to the user. Moreover, in real-world scenarios this information would not even be accessible to the RL agent, as localization of the user induces unreasonable overhead, and furthermore collides with user privacy concerns.

*Figure 11.* Two different viewpoints of the `BeamManagement6G` environment used in this paper. Figure 11(a) shows a single active antenna at every step in time, which corresponds to the task the RL agent has to solve. Figure 11(b) visualizes the underlying ground-truth solution.

In the following subsections, we first detail our model of the radiation-intensity field produced by the base stations in Appendix A.1, followed by a discussion of the relation between beam direction, antenna configuration and codebook element in Appendix A.2. We conclude with the description of the procedure for sampling random trajectories in Appendix A.3.

### A.1. Beam intensity field

We now study the antenna model used in this work and visualized in Figure 12. The base station is equipped with a linear array of antenna elements or senders (orange dots in Figure 12), with the $j$th sender at position $\mathbf{r}_j$. A sender is a dipole radiation source, which we assume to be pointing in perpendicular direction to the 2d plane we consider.

Further, we assume the distance $|\mathbf{r} - \mathbf{r}_j|$ between observer at position $\mathbf{r}$ and the $j$th sender to be much larger than the distance between the individual senders and introduce the effective location $\mathbf{r}_0$ of the antenna such that $\mathbf{r}_j = \mathbf{r}_0 + \mathbf{d}_j$. Below,

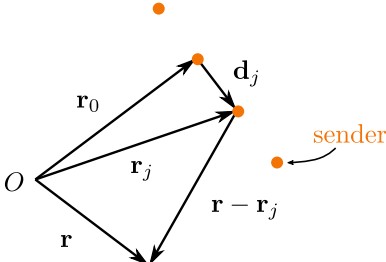

*Figure 12.* Model of an antenna used in the two-dimensional setting used in this work. The senders of a linear phased array are shown as orange dots in the figure. The electric field emanating from the $j$th point source is modeled in the far-field as a spherical wave. A tunable phase offset of each sender can be used to direct beams exploiting interference.

$\mathbf{r}_0$ will point to the sender in the middle of the array (assuming an odd number of senders). Far away from the $j$th sender it's electric field at position $\mathbf{r}$ and time $t$ approximately is a spherical wave (Griffiths, 2013)

$$E_j(\mathbf{r}, t) = \frac{A}{|\mathbf{r} - \mathbf{r}_j|} \sin(k|\mathbf{r} - \mathbf{r}_j| - \omega t + \phi_j). \tag{4}$$

Here, $A$ is a constant, $\omega$ the frequency of the light wave, and $k = \omega/c$ where $c$ is the speed of light is the wave number. We disregard the vector character of the electric field since in the far field of the antenna the electric field component of each sender will approximately point in the same direction. As we will see below, by tuning the phase offset $\phi_j$, directed high-intensity beams can be sent by exploiting interference effects. Next, we define $\mathbf{R} = \mathbf{r} - \mathbf{r}_0$, and $R = |\mathbf{R}|$. The following approximations are valid in the far field, that is whenever $|\mathbf{d}_j|/R \ll 1$ (Griffiths, 2013),

$$|\mathbf{r} - \mathbf{r}_j| = R - \frac{\mathbf{R} \cdot \mathbf{d}_j}{R} + \mathcal{O}(R^{-1}) \tag{5}$$

$$k[|\mathbf{r} - \mathbf{r}_l| - |\mathbf{r} - \mathbf{r}_j|] = k\frac{\mathbf{R} \cdot (\mathbf{d}_j - \mathbf{d}_l)}{R} + \mathcal{O}(R^{-1}) \tag{6}$$

$$\frac{1}{|\mathbf{r} - \mathbf{r}_j|} = \frac{1}{R} + \mathcal{O}(R^{-2}). \tag{7}$$

The average field intensity at a point in space is proportional to the square of the sum of all electric field components averaged over time, that is

$$I(\mathbf{r}) \propto \lim_{T \to \infty} \frac{1}{T} \int_0^T dt \left( \sum_j E_j(t) \right)^2 = \frac{A^2}{2R^2} \left| \sum_j \exp\left\{ i\left( k\frac{\mathbf{R} \cdot \mathbf{d}_j}{R} - \phi_j \right) \right\} \right|^2. \tag{8}$$

The final equality follows by inserting Equation (4), making use of the approximation in Equations (5) to (7), and a subsequent straightforward calculation. We model the base stations as a linear phased array of $N_s$ ($N_s$ odd) senders at position $\mathbf{d}_j = j\mathbf{d}$ with $j = -(N_s - 1)/2, ..., (N_s - 1)/2$ and $\phi_j = j\varphi$. Furthermore, $|\mathbf{d}| = \pi/k$ for maximal constructive interference. Substituting these definitions into Equation (8) yields the final result

$$I = \frac{B^2}{2R^2} \frac{\sin^2(N_s \xi/2)}{\sin^2(\xi/2)} \tag{9}$$

with the proportionality constant $B$ where

$$\xi = k\frac{\mathbf{R} \cdot \mathbf{d}}{R} - \varphi. \tag{10}$$

Since Equation (9) diverges for $R \to 0$, we introduce a cut-off at the value of the intensity at $R = 0.001$. As the number of senders $N_s$ increases, Equation (9) strongly peaks at $\xi = 0$. Measuring the direction of the beam by the angle $\theta$ with respect to $\mathbf{d}$, we find

$$\cos(\theta) = \frac{\varphi}{\pi}. \tag{11}$$

In our simple model, Equation (11) is the mapping between the macroscopic field configuration (the direction $\theta$ of the beam) and the specific physical settings (the phase gradient $\varphi$ of the antenna). The codebook then is a vector $(\theta_1, ..., \theta_{N_c})$ of discretized angles evenly distributed over the interval $[0, 2\pi)$ with the underlying mapping $\varphi_i = \pi \cos(\theta_i)$ with $i = 1, ..., N_c$. In our experiments, we set $N_s = 17$ and $N_c = 9$.

### A.2. Antenna configuration

To generate an instance of the environment, we randomly sample the positions for a predefined number of antennas. Inspired by economic considerations in the real world, we avoid that pairs of antennas are located too close to each other. To enforce this, we first sample antenna coordinates $\mathbf{R}_j$ uniform at random within the $2d$ environment $[0, L] \times [0, L]$. By default, we choose $L = 6$. We resample positions until the Euclidean distance between any of the antennas is greater than a threshold, i.e. if $|\mathbf{R}_i - \mathbf{R}_j| > d_{\min}$ for all $i$ and $j$. The random configurations in our work were created with a value of $d_{\min} = 1.5$. The orientations of the antennas are individually sampled uniformly at random, and successively normalized. For a position $\mathbf{r}$ in the plane the ground truth (optimal antenna and codebook entry) is efficiently calculated by maximizing the intensity at $\mathbf{r}$ by iterating over all codebook elements $\varphi_i$ for each antenna. Figure 13 illustrates some configurations for different numbers of antennas. The intensity field calculated via Equation (9) together with the discretized codebook gives rise to the complex patterns shown in the figure that the RL agent is tasked to learn. Note the difference between the intensity field pattern for a particular codebook element as shown e.g. in Figure 10 and the pattern shown in Figure 13 where the intensity is maximized over the codebook elements independently at each position.

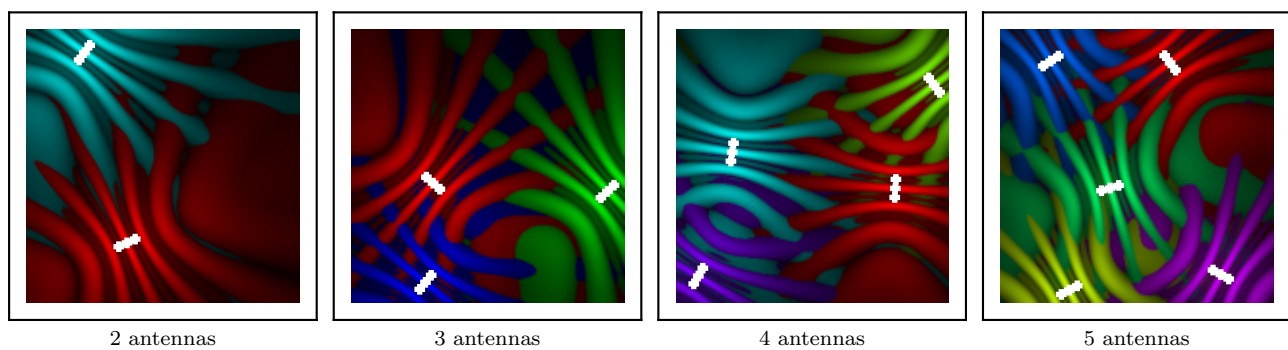

| 2 antennas | 3 antennas | 4 antennas | 5 antennas |

*Figure 13.* Different antenna configurations with increasing number of antennas. At each position, the colors indicate the antenna with maximum intensity after maximization over the codebook elements. The brightness of the colors show the intensity where darker corresponds to lower intensity. For better readability, the plots show $R^2 I$ where $I$ is the antenna's intensity according to Equation (9).

### A.3. Trajectory sampling

In addition to the antenna configuration, we also sample a random trajectory the agent moves on. To this end, we uniformly sample $n \geq 2$ points $(x_i, y_i)$ for $i = 1, ..., n$ within the 2d grid $[0, L] \times [0, L]$. At the boundary we choose $x_1 = 0$ and $x_n = L$. The points are subsequently interpolated by a cubic spline function $\mathbf{s}(t) = (y(t), x(t))$ with $t \in [0, 1]$ and $\mathbf{s}(0) = (0, y_1)$ and $\mathbf{s}(1) = (L, y_n)$. We perform rejection sampling until $0 < y(t) < L$ and $0 < x(t) < L$ for all $t$. The number of support points $n$ allows to control the complexity of the trajectory. Figure 14 exemplarily shows trajectories with an increasing number of support points. Each episode starts at $t = 0$ and ends at $t = 1$. The parametrization of the trajectory $\mathbf{s}(t)$ by $t$ results in a $t$-dependent velocity. As a simplification, we fix the velocity of the agent to $v_0$ which in general is trajectory dependent due to the different arc lengths of the trajectories. To achieve a constant absolute value $v_0$ of the velocity, we re-parametrize $\mathbf{s}(t(\tau))$ and differentiate with respect to $\tau$

$$\frac{\mathrm{d}\mathbf{s}}{\mathrm{d}\tau} = \frac{\partial \mathbf{s}(t)}{\partial t} \frac{\mathrm{d}t}{\mathrm{d}\tau} . \tag{12}$$

Taking the absolute on both sides of the equation, fixing $|\mathrm{d}\mathbf{s}/\mathrm{d}\tau| = v_0$ and abbreviating $v(t) = |\partial \mathbf{s}(t)/\partial t|$, we arrive at the differential equation

$$\frac{\mathrm{d}\tau}{\mathrm{d}t} = \frac{v(t)}{v_0} . \tag{13}$$

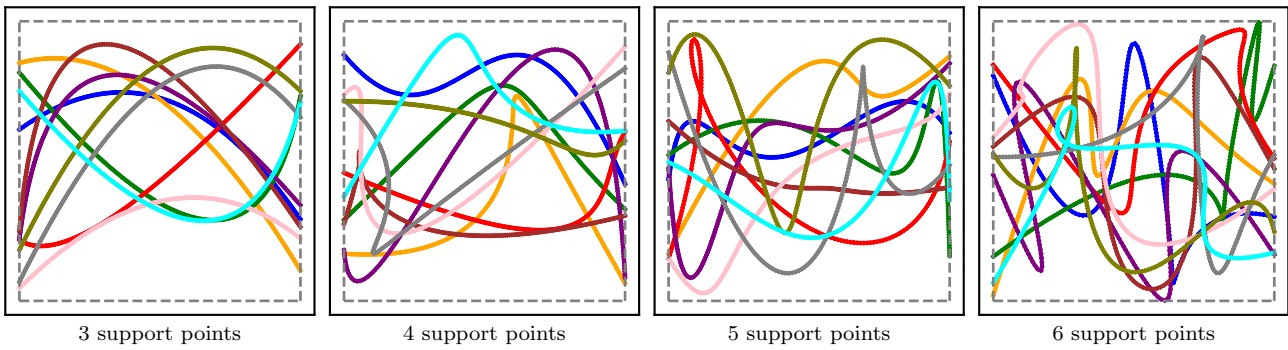

| 3 support points | 4 support points | 5 support points | 6 support points |

*Figure 14.* Different trajectory degrees. We sample $n$ points in the two-dimensional grid and interpolate them by a cubic spline function. The complexity of the trajectories increases with the number of support points $n$. The figure shows sample trajectories for $n = 3, ..., 6$.

Integration with the initial conditions $t(\tau = 0) = 0$ and $t(\tau = 1) = 1$ then yields

$$\tau(t) = \frac{1}{v_0} \int_0^t \mathrm{d}t' v(t') \tag{14}$$

with $v_0 = \int_0^1 \mathrm{d}t' v(t')$. We use this functional dependence between $t$ and $\tau$ to re-parametrize the spline function of the trajectory.

## B. Sample complexity estimator

This appendix provides more details on the statistical estimator for evaluation of sample complexity. Sample complexity, as used in this work, follows the intuitive notion: Sample complexity is the number of interactions with the environment the reinforcement learning algorithm needs to achieve a certain performance. More formally, following Ref. (Kakade, 2003; Kearns & Singh, 1999), given a threshold $V^*$, the sample complexity is the number of interactions with the environment until a measure of the algorithm's performance such as the expected return is larger than the threshold with high probability. This definition is meaningful for RL algorithms with guaranteed monotonic convergence properties to the optimal policy (Kakade, 2003) such as policy iteration in tabular settings (Sutton & Barto, 2018).

However, for algorithms with no performance guarantees, the situation is more involved as showcased by Figure 15. The

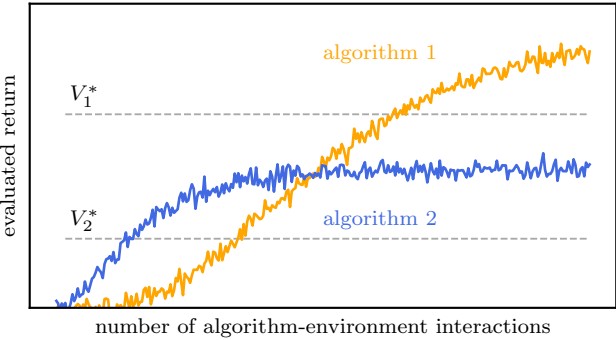

*Figure 15.* The figure shows the learning curves for algorithm 1 and algorithm 2. While algorithm 2 exhibits lower sample complexity with respect to threshold $V_2^*$ than algorithm 1, the converse is true for threshold $V_1^*$, which algorithm 2 even never seems to reach. This example shows the following: If convergence to optimality cannot be proved for the algorithm, the definition of sample complexity is only meaningful when defined with respect to a threshold.

figure exemplarily shows the learning curves of two algorithms. Since we consider here for example neural-network based algorithms with no convergence guarantees to the optimal value, which might even be unknown, the question of which of

these algorithms has lower sample complexity can only be answered with respect to a predefined threshold. Indeed, in Figure 15 algorithm 2 has lower sample complexity with respect to $V_2^*$ than algorithm 1 because the learning curve shown first crosses the threshold $V_2^*$. Conversely, algorithm 1 has lower sample complexity with respect to $V_1^*$ which algorithm 2 even seems to never reach. In this case, we assign infinite sample complexity to algorithm 2.

So far the discussion has been for two specific training runs. In the following, we capture the notion of sample complexity more rigorously by viewing the training curve as the realization of a stochastic process. The stochasticity stems from randomness in the environment but also from the algorithm itself (random initialization of neural network weights, sampling from action distribution, etc.). Statements about sampling complexity under randomness thus require the definition of a statistical estimator introduced in the following:

Consider a random process $\{V_t, t = 1, ..., T\}$. For given $\delta \in (0, 1]$ we define a criterion to decide if $V_t$ is below or above a threshold $V^*$ via the probability $P_t = P(V_t \geq V^*)$ and define the sampling complexity as

$$S = \sum_{t=1}^{T} \mathbb{I}\left[P_t < \delta\right]. \tag{15}$$

Here, $\mathbb{I}[\cdot]$ is the indicator function which is one if its argument is true and zero otherwise. The number of time steps $T$ is sent to infinity but in practice taken to be large enough to capture possible instabilities of the algorithm. Obviously, we do not have access to $P_t$, which we therefore estimate from independent training runs. This leads to the following definition of empirical sample complexity:

**Definition B.1** (Estimator empirical sample complexity – general case)**.** Given a collection of $N$ random processes $\{V_t^{(i)}, t = 1, ..., T\}$ with $i = 1, ..., N$ and $V_t^{(i)}$ i.i.d. for given $t$, a threshold value $V^* \in \mathbb{R}$ and a threshold probability $\delta \in (0, 1]$, we call

$$\hat{S} = \sum_{t=1}^{T} \mathbb{I}\left[\hat{P}_t < \delta\right] \tag{16}$$

where

$$\hat{P}_t = \frac{1}{N} \sum_{i=1}^{N} \mathbb{I}\left[V_t^{(i)} \geq V^*\right], \tag{17}$$

the *empirical sample complexity*.

Note that $\mathbb{E}\hat{P}_t = P_t$, consequently $\hat{P}_t$ is unbiased. The estimator $\hat{S}$ can be simplified when $V_t^{(i)}$ is bounded, e.g. $V_t^{(i)} \in [0, V_{\max}]$. In Section 4 we redefine $V_t^{(i)} \rightarrow V_t^{(i)}/V_{\max}$ and $V^* \rightarrow V^*/V_{\max} := 1 - \varepsilon$ with $\varepsilon \in [0, 1]$. In this case Equation (17) becomes

$$\hat{P}_t = \frac{1}{N} \sum_{i=1}^{N} \mathbb{I}\left[V_t^{(i)} \geq 1 - \varepsilon\right]. \tag{18}$$

In the following, we investigate the properties of $\hat{S}$, in particular consistency and bias.

**Theorem B.2** (Consistency)**.** *$\hat{S}$ is consistent, that is for all $\eta > 0$ we find that $\lim_{N \to \infty} P\left(|\hat{S} - S| > \eta\right) = 0$*

*Proof.*

$$P\left(|\hat{S} - S| > \eta\right) \leq \frac{1}{\eta} \mathbb{E}|\hat{S} - S| \tag{19}$$

$$\leq \frac{1}{\eta} \sum_{t=1}^{T} \mathbb{E}\left|\mathbb{I}[\hat{P}_t < \delta] - \mathbb{I}[P_t < \delta]\right| \tag{20}$$

$$= \frac{1}{\eta} \sum_{t=1}^{T} \left|P(\hat{P}_t < \delta) - \mathbb{I}[P_t < \delta]\right|. \tag{21}$$

We first used Markov's inequality, followed by the triangle inequality. In the final step we made use of the fact that the indicator function is either zero or one and that $\mathbb{E}\,\mathbb{I}[\cdot] = P(\cdot)$. With the definition $\Delta = \delta - P_t$, we find

$$P\big(\hat{P}_t < \delta\big) = P\big(\hat{P}_t - P_t < \Delta\big) = \mathbb{I}[P_t < \delta] + \begin{cases} P\big(\hat{P}_t - P_t < -|\Delta|\big) & \text{if } \Delta \le 0 \\ -P\big(\hat{P}_t - P_t \ge \Delta\big) & \text{if } \Delta > 0 \end{cases} \tag{22}$$

and thus

$$\lim_{N\to\infty} P\big(\hat{P}_t < \delta\big) = \mathbb{I}[P_t < \delta], \tag{23}$$

recalling that $\hat{P}_t$ is a consistent estimator for $P_t$, that is $\lim_{N\to\infty} P\big(|\hat{P}_t - P_t| > \Delta\big) = 0$ for $\Delta > 0$. The claim then follows when taking the limit of Equation (19) using Equation (23). $\qquad\square$

Let us now consider the large $N$ limit and possible bias of the estimator. To this end, we calculate the probability of $\mathbb{I}[\hat{P}_t < \delta] = 1$ even though $P_t > \delta$.

**Theorem B.3** (Bias). *For large $N$ we have $P(\mathbb{I}[\hat{P}_t < \delta] = 1) \to \Phi(\Delta\sqrt{N}/\sigma)$ with $\Delta = \delta - P_t$ and $\sigma = \sqrt{P_t(1-P_t)}$ where $\Phi$ is the error function.*

*Proof.* We obtain an expression for $P(\mathbb{I}[\hat{P}_t < \delta] = 1)$ by the central-limit theorem. We find

$$P(\mathbb{I}[\hat{P}_t < \delta] = 1) = P(\hat{P}_t < \delta) \to \Phi(\Delta\sqrt{N}/\sigma) \quad (N \text{ large}) \tag{24}$$

where $\sigma^2 = \mathrm{Var}\big(\mathbb{I}[V_t^{(i)} \ge V^*]\big) = P_t(1-P_t)$ and the error function $\Phi(z) = \frac{1}{\sqrt{2\pi}} \int_{-\infty}^{z} \mathrm{d}z\, e^{-\frac{1}{2}z^2}$. $\qquad\square$

The estimator therefore is slightly biased around $P_t \approx \delta$ where Equation (24) deviates from the step function $\mathbb{I}(P_t < \delta)$ over a width of $\sqrt{\delta(1-\delta)}/\sqrt{N}$. Figure 16 shows the numerical evaluation for $N = 100$ samples and $V^{(i)} \sim \mathrm{Uniform}(0,1)$. The smoothed step function is clearly visible for different values of $\delta$. For the uniformly distributed random variables considered here, the central limit theorem closely approximates the simulated curves for $N = 100$ which corresponds to the chosen number of training runs throughout this work.

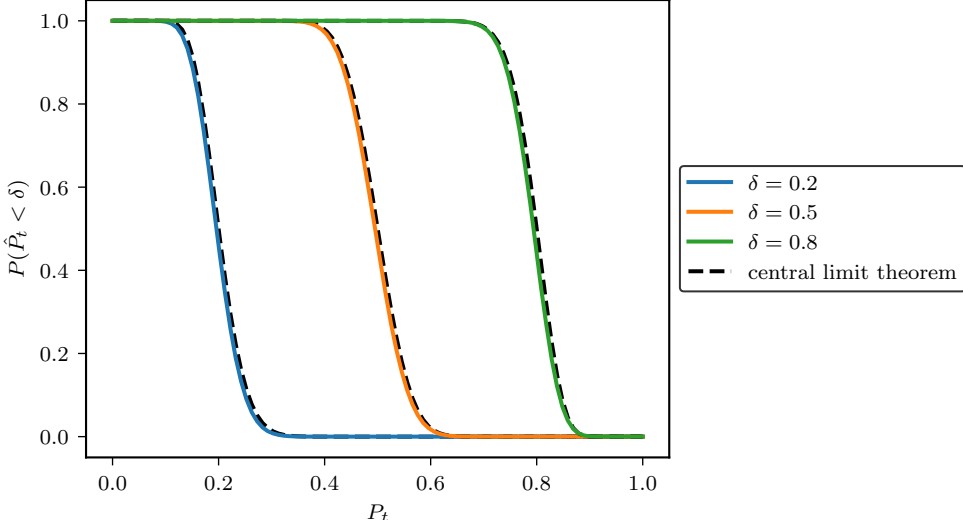

*Figure 16.* The figure shows numerical simulations for $V^{(i)} \sim \mathrm{Uniform}(0,1)$. The $y$ axis shows the probability $P(\mathbb{I}[\hat{P}_t < \delta] = 1) = P(\hat{P}_t < \delta)$, on the $x$-axis we scan $P_t = 1 - V^*$. The figure shows smoothed step functions around $P_t = \delta$. For the exemplary distribution $V^{(i)} \sim \mathrm{Uniform}(0,1)$, the central limit theorem (dashed lines) approximates the functions well.

## C. Experimental Details

This appendix provides details on the experimental setup that was used for producing the results in this paper. Moreover, we will highlight the implemented tools and configuration options. All implementations and raw results are accessible as described in the data availability statement.

We implemented the pipeline in `python` to be compatible with most of the ongoing research efforts in the machine learning (ML) and quantum computing (QC) community. The classical routines are mainly based on the `PyTorch` library (Paszke et al., 2019). For hyperparameter optimization of the quantum models we made use of the `qiskit-torch-module` (Meyer et al., 2024), a library for fast simulation of quantum neural networks on multi-core systems. These initial experiments (results see Appendix C.1) were conducted on a system with a AMD Ryzen 9 5900X 12-Core CPU. As the successive experiments exceeded the capacity of a single machine, these were executed on the `woody` cluster of the Erlangen National Performance Computing Center (NHR@FAU), consisting of 112 nodes of Intel Xeon E3-1240 v6 4-Core CPUs. As these are addressable in a single-core granularity, the `PennyLane` library (Bergholm et al., 2022) was found to be more efficient for simulating the quantum circuits.

The pipeline itself consists of three main components:

(1) A realization of the `BeamManagement6G` environment described in Appendix A, following the OpenAI `gymnasium` API (Brockman et al., 2016): The three-dimensional observation consists of base station index, received intensity, and codebook element of the previous time step. The first and last elements are normalized component-wise to the range $[0, 1]$. The intensity is inherently restrained to $[0, 1]$. The implementation allows to stack multiple past observations to provide more information to the agent, but this was not found to improve the overall performance for the setups considered in this paper. During training, the absolute received intensity value is returned as reward value. For testing purposes, it is also possible to report the fraction of the maximum achievable intensity. However, this internally computes the ground-truth solution, and therefore this information should not be used during training phase.

The agent's action is to select one of the available base stations by its index. Once selected, the environment internally performs a sweep over the available codebook elements and selects that corresponding to the greatest intensity at the position of the UE. The internal spatial position of the environment is updated according to a trajectory object following Appendix A.3. For the experiments in this paper, we selected a constant trajectory length of 200 steps. If not mentioned otherwise, the experiments were conducted with the three-antenna setup displayed in Figure 11. Our framework is implemented in a modular fashion, which allows to easily replace this environment with others available through the `gymnasium` library.

(2) A RL algorithm that trains on the environment for a user-defined maximum number of steps. For that purpose, we realized two different algorithms: Double Deep Q-Networks (DDQN) (Van Hasselt et al., 2016), an off-policy value-function based routine; Proximal Policy Optimization (PPO) (Schulman et al., 2017), an on-policy policy-function based routine. Most experiments in this paper refer to the DDQN algorithm, as off-policy algorithms are usually more sample-efficient than on-policy approaches (Sutton & Barto, 2018). However, to provide an additional example of using our sample complexity estimator, we added some experimental results for PPO in Appendix C.2. Both algorithms are integrated from the `Tianshou` library (Weng et al., 2022), which makes it straightforward to extend our framework with additional RL algorithms. Our framework implements classical neural networks and hybrid quantum-classical networks as function approximators. Details on configuration possibilities and model sizes are provided and analyzed in Appendix D.

During the training procedure, validation is performed in regular intervals. The training and validation trajectories and all associated rewards are logged for successive computation of the sampling complexity. Additionally, intermediate and final parameters of the model are stored, which allows for retrospective fine-tuning and testing.

(3) A tool for evaluating the sample complexity, based on the logged trajectories from the training routine. Additionally, we implement the estimation of percentiles via cluster re-sampling (Cameron et al., 2008). This is realized as a post-processing step and requires only few computational resources once the training data is available.

A pseudocode overview of the end-to-end pipeline for estimating the sample complexity of DDQN can be found in Algorithm 1. A modified version for the PPO algorithm is provided in Algorithm 2.

---

**Algorithm 1** Estimating the Sample Complexity of Double Deep Q-Networks (DDQN)

---

**Input:** environment $\mathcal{E}$ (with a horizon of 200 for `BeamManagement6G`), state-action value function approximator $Q$,
**Input:** training runs $N_{\text{seed}}$ (defaults to 100), epochs to train for $N_{\text{epoch}}$ (defaults to 100),
**Input:** number of training environments $N_{\text{env}}$ (defaults to 10), validation environments $N_{\text{val}}$ (defaults to 100)
**Output:** sample complexity of DDQN on environment $\mathcal{E}$ for threshold probabilities $\delta_0, \ldots$ and error thresholds $\varepsilon_0, \ldots$

Set up empty reward_buffer of shape $N_{\text{seed}} \times N_{\text{epoch}}$
**for** $N_{\text{seed}}$ different (random) initial parametrizations of $Q$ **do**
    Initialize a standard DDQN algorithm (Van Hasselt et al., 2016; Weng et al., 2022) with $Q$ and hyperparameters
    $-\ \varepsilon_{\text{greedy}}$: epsilon-greedy action selection
    $-\ \alpha_C, \alpha_Q$: classical and (optional) quantum learning rate
    $-\ \gamma$: reward discount factor
    $-\ N_{\text{sync}}$: target network synchronization rate
    $-\ N_{\text{buffer}}$: experience replay buffer size
    $-\ N_{\text{batch}}$: mini-batch size for update
    **for** $N_{\text{epoch}}$ training epochs **do**
        Use the DDQN algorithm on $N_{\text{env}}$ parallel training environments
        Employ current policy on $N_{\text{val}}$ parallel validation environments
        Store the averaged validation result to the reward_buffer
    **end for**
**end for**
**for** threshold probabilities $\delta_0, \delta_1, \ldots$ **do**
    **for** error thresholds $\varepsilon_0, \varepsilon_1, \ldots$ **do**
        Determine the sample complexity and respective percentiles from the reward_buffer as described in Appendix B
    **end for**
**end for**

---

---

**Algorithm 2** Estimating the Sample Complexity of Proximal Policy Optimization (PPO)

---

**Input:** environment $\mathcal{E}$ (with a horizon of 200 for `BeamManagement6G`), actor approximator $\Pi$, critic approximator $V$,
**Input:** training runs $N_{\text{seed}}$ (defaults to 100), epochs to train for $N_{\text{epoch}}$ (defaults to 500),
**Input:** number of training environments $N_{\text{env}}$ (defaults to 10), validation environments $N_{\text{val}}$ (defaults to 100)
**Output:** sample complexity of PPO on environment $\mathcal{E}$ for threshold probabilities $\delta_0, \ldots$ and error thresholds $\varepsilon_0, \ldots$

Set up empty reward_buffer of shape $N_{\text{seed}} \times N_{\text{epoch}}$
**for** $N_{\text{seed}}$ different (random) initial parametrizations of $\Pi, V$ **do**
    Initialize a standard PPO algorithm (Schulman et al., 2017; Weng et al., 2022) with $\Pi, V$ and hyperparameters
    $-\ \varepsilon_{\text{clip}}$: gradient clipping threshold
    $-\ \alpha_C, \alpha_Q$: classical and (optional) quantum learning rate
    $-\ \gamma$: reward discount factor
    $-\ N_{\text{batch}}$: mini-batch size for update
    $-\ N_{\text{repeat}}$: update repetition for each batch
    **for** $N_{\text{epoch}}$ training epochs **do**
        Use the PPO algorithm on $N_{\text{env}}$ parallel training environments
        Employ current policy on $N_{\text{val}}$ parallel validation environments
        Store the averaged validation result to the reward_buffer
    **end for**
**end for**
**for** threshold probabilities $\delta_0, \delta_1, \ldots$ **do**
    **for** error thresholds $\varepsilon_0, \varepsilon_1, \ldots$ **do**
        Determine the sample complexity and respective percentiles from the reward_buffer as described in Appendix B
    **end for**
**end for**

---

## C.1. Hyperparameter Optimization

To evaluate and compare the sample efficiency, it is important to optimize the hyperparameters of the involved algorithms. Only then statements can be made on the suitability and performance for specific tasks. While we focus on algorithmic hyperparameters in this section, an ablation study for different underlying models is performed in Appendix D. While it is impossible to consider every degree of freedom with fine granularity, we determine the hyperparameters that have the largest impact on the overall performance – and in particular sample complexity.

For both the DDQN and PPO algorithm with classical and quantum function approximators the results are reported in Table 1. The bold values were determined to be optimal by using grid-search with 20 runs for each configuration. These are the settings that have been used to produce the results in the rest of this paper. While this does not proof that the respective models produce the best sample efficiency possible, we believe that the performed comprehensive hyperparamter optimization allows for statements with high certainty.

| | | **Classical** | **Quantum** |
|---|---|---|---|
| DDQN (Algorithm 1) | action selection $\varepsilon_{\text{greedy}}$ | 0.05, **0.1**, 0.2 | 0.05, **0.1**, 0.2 |
| | synchronization rate $N_{\text{sync}}$ | 250, **1000**, 2000, 4000 | not separately optimized |
| | replay buffer size $N_{\text{buffer}}$ | **1000**, 10000 | not separately optimized |
| | classical learning rate $\alpha_C$ | 0.01, 0.001, **0.0005**, 0.0002, 0.0001 | 0.001, **0.0005**, 0.0002, 0.0001 |
| | quantum learning rate $\alpha_Q$ | N/A | 0.002, **0.001**, 0.0005, 0.0002 |
| PPO (Algorithm 2) | gradient clipping $\varepsilon_{\text{clip}}$ | 0.05, **0.1**, 0.2, 0.4 | 0.05, 0.1, **0.2**, 0.4 |
| | update repetition $N_{\text{repeat}}$ | 1, 5, **10**, 50, 100 | not separately optimized |
| | classical learning rate $\alpha_C$ | 0.002, **0.001**, 0.0005, 0.0002, 0.0001 | 0.002, **0.001**, 0.0005 |
| | quantum learning rate $\alpha_Q$ | N/A | 0.002, **0.001**, 0.0005 |
| Shared | discount factor $\gamma$ | 0.90, **0.95**, 0.99 | not separately optimized |
| | mini-batch size $N_{\text{batch}}$ | 32, **64** | not separately optimized |
| | activation function[†] | None, **ReLU**, Tanh | **None**, ReLU, Tanh |

[†]refers to the type of activation function used in classical neural networks between each layer; for hybrid classical-quantum networks, for a sketch see Figure 3, placement is between input layer and quantum circuit, as well as between measurement and output layer.

*Table 1.* Hyperparameter optimization for the DDQN and PPO algorithm with underlying classical and quantum models. For each hyperparameter, we denote the considered values and the found optimal setup. For guaranteeing robust results, we performed full grid search with 20 seeds for each configuration. Due to the large simulation overhead, for the quantum models some hyperparameters were not separately optimized but taken over from the classical results.

## C.2. Sampling Complexity of Proximal Policy Optimization (PPO)

Most experiments in this paper were conducted using the DDQN algorithm (Van Hasselt et al., 2016), which can be considered an off-policy approach. This means that the agent can learn from actions that were not produced by the current policy. In DDQN this is typically realized by an experience replay buffer that allows to re-use past experience (Mnih et al., 2015). It is straightforward to see that such re-use of information can be expected to reduce the sample complexity. On the other hand, PPO (Schulman et al., 2017) is an on-policy algorithm. Such approaches can only learn from actions that were taken during the current training epoch. While this allows for more stable learning in some contexts, as well as other advantages (Sutton & Barto, 2018), it does prevent the use of past information. Consequently, the sample efficiency can be expected to be tentatively lower compared to DDQN. However, it is still possible to quantitatively compare PPO with different underlying models.

In Figure 17, we report the sample complexities for three different classical PPO models and one quantum PPO model on the `BeamManagement6G` environment. To allow for convergence, we increased the maximum number of interactions to 500 (epochs) · 200 (steps) · 10 (batch) = 1000000. Comparison with Figure 6 validate the above assumption that PPO is less sample efficient than DDQN. However, for most $\varepsilon$-$\delta$-configurations we observe superior performance of the small quantum model to the small classical model (width 16), on par performance with the medium classical model (width 32), and only slightly inferior performance to the large classical model (width 64). This is in line with the other observations in this work, i.e. that quantum models can achieve a sample complexity competitive with much larger classical models.

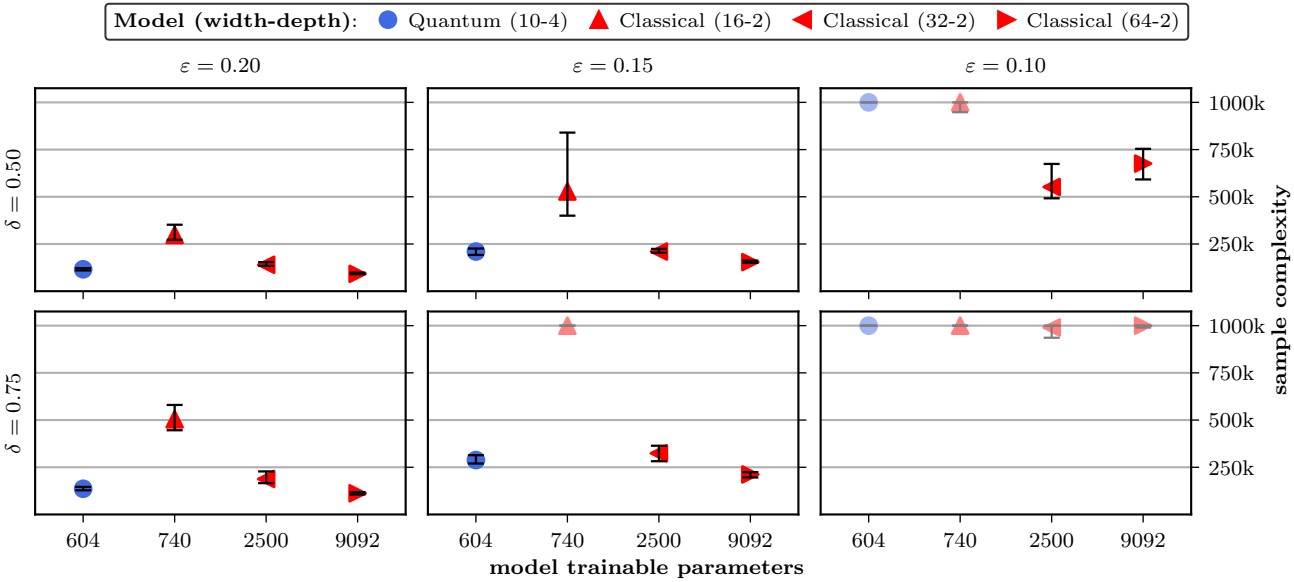

*Figure 17.* Empirical sample complexities $\hat{S}$ of proximal policy optimization (PPO) for various relative errors $\varepsilon$ and threshold probabilities $\delta$ on the `BeamManagement6G` environment. The model width denotes the number of neurons in classical hidden layers, and the number of qubits in quantum models, respectively. Moreover, model depth refers to the number of hidden layers in DNNs, and number of ansatz repetitions in VQCs. Compared to Figure 6 which shows the same study but for DDQN, the larger number of trainable parameter originates from the use of separate actor and critic networks. All results are averaged over 100 seeds and error bars denote the 5th and 95th percentiles, estimated with cluster resampling. Configurations where multiple runs do not achieve the targeted relative error rate before the cut-off of one million steps are grayed out.

## D. Analysis of Classical and Quantum Function Approximators

This appendix complements the discussion of the algorithmic setup from Appendix C with considerations regarding the underlying models. In our work, we distinguished between two classes of function approximators – both for the value function in DDQN and the policy in PPO: On the one hand, classical fully-connected neural networks with varying *width* and *depth* of the hidden layers. On the other hand, hybrid classical-quantum networks, i.e. a variational quantum circuit with varying *qubit* and *layer* counts, encased by single-layer classical networks (sketch see Figure 3).

In Appendix D.1, we describe the hybrid model in more detail and discuss different ansätze for the underlying variational quantum circuit. An ablation study we conducted w.r.t. model complexity of both, classical and quantum approaches, is outlined in Appendix D.2.

### D.1. Notes on Quantum Circuit Ansatz

The choice of the quantum circuit ansatz is a frequently debated topic in the quantum computing community. It is possible to compare different choices based on measures such as expressibility, trainability, and entanglement capability (Sim et al., 2019; Abbas et al., 2021). There also exist approaches for automatically generating architectures that are optimal w.r.t. some of these properties (Du et al., 2022). However, apart from some artificial examples, relating these metrics to task-specific performance has so far been unsuccessful.

We decided for a hybrid instead of a pure quantum model which avoids the dependency of the qubit number on the dimensionality of the RL state. This allows the flexibility of arbitrarily scaling the VQC – for a sketch see Figure 3. More formally, we use an initial single-layer fully connected network to map the dimension of the state observation $\mathbf{s} := \mathbf{s}^{(0)}$ to a vector compatible with the input dimension of an $n$-qubit VQC:

$$\mathbf{s}^{(1)} = \mathbf{w}_{\text{pre}}^t \cdot \mathbf{s}^{(0)} + \mathbf{b}_{\text{pre}} \tag{25}$$

This intermediate state $\mathbf{s}^{(1)}$ is encoded into the VQC with a parameterized unitary $U(\mathbf{s}^{(1)}; \Theta)$, with details on the trainable

parameters $\Theta$ and encoding procedure described below. The respective quantum state is evolved and the individual qubits are measured in the Pauli-Z basis to get the intermediate state $\mathbf{s}^{(2)}$:

$$\mathbf{s}^{(2)} = \begin{bmatrix} \langle 0| U(\boldsymbol{s}^{(1)};\Theta)^{\dagger} \left( Z \otimes I^{\otimes n-1} \right) U(\boldsymbol{s}^{(1)};\Theta)|0\rangle \\ \vdots \\ \langle 0| U(\boldsymbol{s}^{(1)};\Theta)^{\dagger} \left( I^{\otimes n-1} \otimes Z \right) U(\boldsymbol{s}^{(1)};\Theta)|0\rangle \end{bmatrix} \tag{26}$$

The intermediate result is post-processed using another single-layer classical neural network. This adjusts the dimensionality to the desired output dimension, usually the number of actions, i.e. antennas in the `BeamManagement6G` environment:

$$\mathbf{s}^{(3)} = \mathbf{w}_{\text{post}}^{t} \cdot \mathbf{s}^{(2)} + \mathbf{b}_{\text{post}} \tag{27}$$

For the DDQN algorithm, this output $\mathbf{s}^{(3)}$ is directly used as an approximation for the Q-value function. In case of PPO, we consecutively append a softmax layer to get a probability density function approximating the policy. While the hybrid model incorporates both, trainable classical and quantum parameters, our ablation study on model complexity in Appendix D.2 demonstrates that the performance of these models can be mainly attributed to the quantum part.

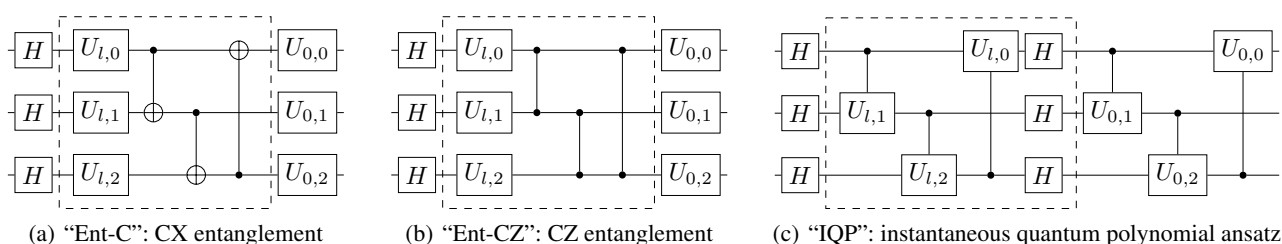

(a) "Ent-C": CX entanglement      (b) "Ent-CZ": CZ entanglement      (c) "IQP": instantaneous quantum polynomial ansatz

*Figure 18.* Three different hardware-efficient ansatz structures compared for this paper. All configurations initially create a uniform superposition, followed by potentially multiple layers of variational gates and two-qubit entangling gates. The different instances are realized with: (a) variational single-qubit unitaries together with nearest-neighbor controlled-X gates; (b) variational single-qubit unitaries together with nearest-neighbor controlled-Z gates; (c) controlled variational unitaries in a nearest-neighbor structure, followed by Hadamard gates; In the notation $U_{l,q}$, the index $l$ denotes the layer and $q$ the qubit position. Consequently, for two and more layers we employ data re-uploading (Pérez-Salinas et al., 2020), enhancing the expressivity of the ansatz. The variational unitaries are realized with one of the parameterizations of a universal single-qubit rotation shown below in Equations (29) to (31).

For studying the measure of sample complexity, we identified the most suitable ansatz out of 9 architectures that are commonly used by the quantum machine learning (QML) community. More concretely, we consider 3 different hardware-efficient circuit structures, all featuring at most two-qubit interactions (Kandala et al., 2017). This includes (a) a nearest-neighbor entanglement structure with controlled-X gates and single-qubit rotations, (b) a nearest-neighbor entanglement structure with controlled-Z gates and single-qubit rotations, and (c) the so-called *instantaneous quantum polynomial* ansatz (Shepherd & Bremner, 2009) with controlled rotations interleaved between Hadamard gates. A visualization of these configurations can be found in Figure 18. In general, the single-qubit unitary $U_{l,q}(\mathbf{s};\Theta)$ acts on qubit $q$ in layer $l$. We realize data encoding in the general form as

$$U_{l,q}(\mathbf{s};\theta,\lambda) = U(\lambda_{l,q,2} \cdot s_q + \theta_{l,q,2},\ \lambda_{l,q,1} \cdot s_q + \theta_{l,q,1},\ \lambda_{l,q,0} \cdot s_q + \theta_{l,q,0}), \tag{28}$$

where the trainable parameters $\Theta$ are comprised of *standard* variational parameters $\theta$, and state scaling parameters $\lambda$ (Jerbi et al., 2021). Furthermore, $s_q$ denotes the $q$-th entry of the intermediate output from the classical encoding layer, see Equation (25). In this work, we consider 3 different parameterization of variational universal rotation gates:

$$U_{l,q}^{\text{ROT}}(\mathbf{s};\theta,\lambda) = R_z(\lambda_{l,q,2} \cdot s_q + \theta_{l,q,2}) R_y(\lambda_{l,q,1} \cdot s_q + \theta_{l,q,1}) R_z(\lambda_{l,q,0} \cdot s_q + \theta_{l,q,0}) \tag{29}$$

$$U_{l,q}^{\text{XYZ}}(\mathbf{s};\theta,\lambda) = R_z(\lambda_{l,q,2} \cdot s_q + \theta_{l,q,2}) R_y(\lambda_{l,q,1} \cdot s_q + \theta_{l,q,1}) R_x(\lambda_{l,q,0} \cdot s_q + \theta_{l,q,0}) \tag{30}$$

$$U_{l,q}^{\text{U3}}(\mathbf{s};\theta,\lambda) = U_3(\lambda_{l,q,2} \cdot s_q + \theta_{l,q,2},\ \lambda_{l,q,1} \cdot s_q + \theta_{l,q,1},\ \lambda_{l,q,0} \cdot s_q + \theta_{l,q,0}) \tag{31}$$

All three expressions parametrize arbitrary single-qubit unitaries. The representation in Equation (29) is typically used by `PennyLane`. The second on in Equation (30) is the subsequent rotation along all three axis. The final one in Equation (31)

| gate | | structure | | |
|---|---|---|---|---|
| | | IQP | Ent-CX | Ent-CZ |
| | ROT | baseline throughout paper | ⊖ slightly worse performance | ⊖ significantly worse performance |
| | XYZ | ⊖ slightly worse performance 
 ⊖ slow simulation | ⊕ comparable performance 
 ⊖ slow simulation | ⊖ significantly worse performance 
 ⊖ slow simulation |
| | U3 | ⊖ extremely slow simulation 
 ⊖ unstable training curves | ⊖ unstable training curves | ⊖ significantly worse performance 
 ⊖ unstable training curves |

*Table 2.* Different quantum circuit ansätze compared to the baseline selected for the experiments in this paper. We considered combinations of the ansatz layouts in Figure 18 and variational gates in Equations (29) to (31). Overall we observed that the IQP and CX-entanglement layout exhibited a performance clearly superior to CZ-entanglement. Furthermore, ROT and XYZ gates performed similarly, but the former was much faster to simulate. While the final performance using U3 gates was also comparable, the training procedure was much more volatile.

is the arbitrary single-qubit rotation parameterized as

$$U_3(\theta, \phi, \delta) = \begin{bmatrix} \cos\left(\frac{\theta}{2}\right) & -e^{i\delta}\sin\left(\frac{\theta}{2}\right) \\ e^{i\phi}\sin\left(\frac{\theta}{2}\right) & e^{i(\phi+\delta)}\cos\left(\frac{\theta}{2}\right) \end{bmatrix}. \tag{32}$$

While in principle all these parameterizations can be used to approximate the same functions, we observed significant differences regarding convergence stability and simulation speed. Our results are summarized in Table 2. Overall, we identified the IQP structure with $U^{\mathrm{ROT}}$ parameterization as most suitable for optimizing the sample complexity on the `BeamManagement6G` task. Therefore, this configuration is used for all other experiments in this paper. The combination of the CX-Ent structure with $U^{\mathrm{XYZ}}$ parameterization exhibits almost equivalent performance but increased the simulation times due to not being native in `PennyLane`. As a general rule of thumb, we discovered that the IQP and CX-Ent structure are superior to CZ-Ent. Moreover, the training performance with $U^{\mathrm{ROT}}$ and $U^{\mathrm{XYZ}}$ usually was much more stable compared to $U^{\mathrm{U3}}$. We emphasize that this small study by no means should be considered an exhaustive architecture search. It might be possible to develop ansätze that are even more suitable for the considered task. However, such extensions are out of the scope of this work.

### D.2. Ablation Study of Model Complexity

In the following, we will investigate the impact of model complexity on the sample complexity. For both types of models we examined two degrees of freedom: For the classical model, this incorporates the *depth*, i.e. number of hidden layers, and the *width*, i.e. the number of neurons in each hidden layer. With input dimension $\dim_{\mathrm{in}}$ and output dimension $\dim_{\mathrm{out}}$, the number of trainable parameters scales as:

$$\text{weight parameters} : \overbrace{\dim_{\mathrm{in}} \cdot \mathrm{width}}^{\text{input layer}} + \overbrace{(\mathrm{depth} - 1) \cdot \mathrm{width} \cdot \mathrm{width}}^{\text{hidden layer(s)}} + \overbrace{\mathrm{width} \cdot \dim_{\mathrm{out}}}^{\text{output layer}} \tag{33}$$

$$\text{bias parameters} : \mathrm{width} + (\mathrm{depth} - 1) \cdot \mathrm{width} + \dim_{\mathrm{out}} \tag{34}$$

For the standard `BeamManagement6G` environment configuration with $\dim_{\mathrm{in}} = 3$ (i.e. antenna index, received intensity, and codebook element of previous timestep) and $\dim_{\mathrm{out}} = 3$ (i.e. 3 antennas) this simplifies to

$$(\mathrm{depth} - 1) \cdot \mathrm{width}^2 + (\mathrm{depth} + 6) \cdot \mathrm{width} + 3 \tag{35}$$

overall trainable parameters, i.e. a linear scaling in the depth and quadratic in the width. The parameter count approximately doubles for PPO as separate actor and critic networks are used – with the critic requiring an output size of 1.

For the hybrid classical-quantum model, the number of classical parameters depends on the in- and output dimensionality, as well as on the number of *qubits* in the VQC. Additionally, we can increase the number of variational parameters by

appending additional *layers*. Therefore, the number of trainable parameters scales as:

$$\text{(classical) weight parameters :} \quad \overbrace{\dim_{\text{in}} \cdot \text{qubits}}^{\text{input layer}} + 0 + \overbrace{\text{qubits} \cdot \dim_{\text{out}}}^{\text{output layer}} \quad (36)$$

$$\text{(classical) bias parameters :} \quad \text{qubits} + 0 + \dim_{\text{out}} \quad (37)$$

$$\text{(quantum) variational parameters :} \quad 0 + \text{layers} \cdot \text{qubits} \cdot 3 + 0 \quad (38)$$

$$\text{(quantum) scaling parameters :} \quad 0 + \underbrace{\text{layers} \cdot \text{qubits} \cdot 3}_{\text{quantum circuit}} + 0 \quad (39)$$

For the standard environment configuration with this simplifies to

$$\underbrace{6 \cdot \text{layers} \cdot \text{qubits}}_{\text{quantum parameters}} + \underbrace{7 \cdot \text{qubits} + 3}_{\text{classical parameters}} \quad (40)$$

overall trainable parameters. One can see from this expression that for increasing number of layers, the fraction of trainable classical parameters becomes insignificant in comparison to the parameters count of the quantum circuit. This ensures, that the performance of the model originates from the quantum part, while only pre- and post-processing is handled classically. Moreover, further below we show that small purely classical networks are not capable of learning a meaningful strategy in the `BeamManagement6G` environment.

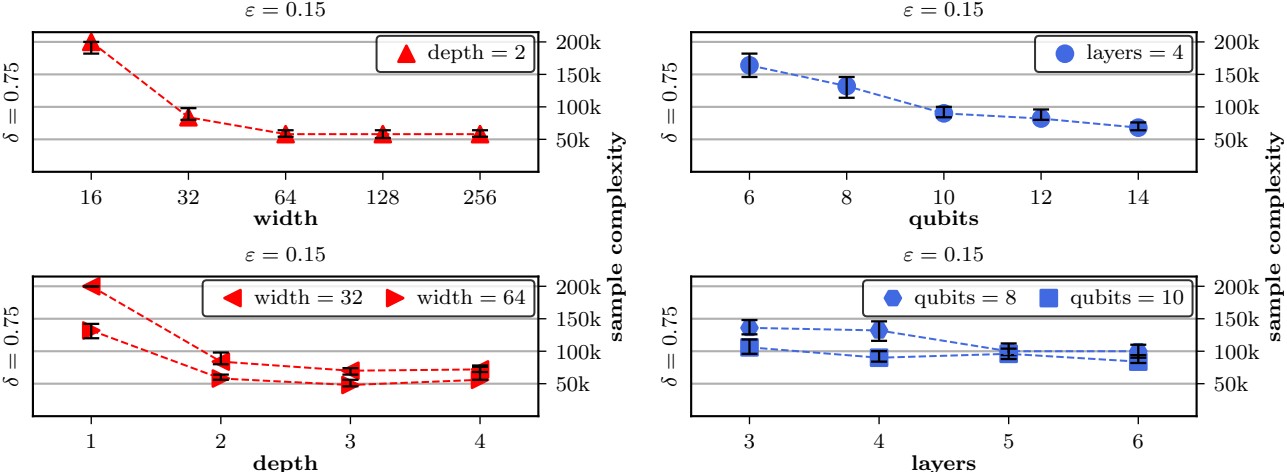

(a) Classical neural networks with: (upper plot) increasing width of hidden layers for depth 2; (lower plot) increasing depth of hidden layers for widths 32 and 64.

(b) Hybrid classical-quantum networks with: (upper plot) increasing number of qubits for 4 variational layers; (lower plot) increasing number of variational layers for 8 and 10 qubits.

*Figure 19.* Impact of model size on the sample complexity in the `BeamManagement6G` environment with the DDQN algorithm. For simplicity, we show only a single intermediate configuration of threshold probability $\delta = 0.85$ and error threshold $\varepsilon = 0.15$, but similar results were observed for other setups. In (a) we depict the scaling behavior of a classical model with increasing width and depth, (b) refers to a quantum model with increasing qubit count and number of layers. The concrete parameter counts can be determined following Equations (35) and (40). All results are averaged over 100 seeds, and error bars denote the 5th and 95th percentiles, estimated with 1000 repetitions of cluster bootstrapping.

We analyze the impact of model complexity, in terms of the trainable parameter count, on the sample complexity in Figure 19. All experiments were conducted on the `BeamManagement6G` environment from Figure 11, using the DDQN algorithm. We report results for an in-between configuration of threshold probability $\delta = 0.85$ and error threshold $\varepsilon = 0.15$.

For the classical models in Figure 19(a), a clear performance saturation with model complexity can be observed. When the number of hidden layers is fixed to 2, the best sample complexity is achieved with a hidden layer width of 64 (4611 parameters). The performance with a width of 128 (17411 parameters) and 256 (67587 parameters) is nearly equivalent but no improvement could be observed. Consequently, there should be a sweet-spot model size for optimizing the task-specific

| | | | qubits | | | | |
|---|---|---|---|---|---|---|---|
| | | | 6 | 8 | 10 | 12 | 14 |
| **layers** | 3 | total runtime [min:sec] | | 00:48 | 01:19 | | |
| | | thereof for train \| test | | 00:23 \| 00:19 | 00:44 \| 00:26 | | |
| | 4 | total runtime [min:sec] | 00:44 | 00:59 | 01:35 | 04:10 | 12:41 |
| | | thereof for train \| test | 00:20 \| 00:18 | 00:30 \| 00:21 | 00:55 \| 00:29 | 02:46 \| 01:07 | 09:21 \| 02:48 |
| | 5 | total runtime [min:sec] | | 01:09 | 01:57 | | |
| | | thereof for train \| test | | 00:37 \| 00:23 | 01:10 \| 00:33 | | |
| | 6 | total runtime [min:sec] | | 01:57 | 02:17 | | |
| | | thereof for train \| test | | 01:10 \| 00:33 | 01:24 \| 00:37 | | |

*Table 3.* Expected runtimes for simulating one epoch of DDQN training with hybrid classical-quantum models. All times refer to single-core performance on an Intel Xeon E3-1240 v6 CPU. Keep in mind that for the results in this work we typically trained for 100 epochs and averaged the performance over 100 runs. We report the *total* end-to-end times, as well as two other values: the time required for actual *train*ing, mostly for computing the gradients – realized using the `backprop` method from `PennyLane`; the time for intermediate testing, which is necessary for subsequent estimation of the sample complexity; For the classical models of the considered sizes, the time was approximately constant at only 15 seconds per episode.

sample complexity. A similar observation is made for increasing model depth, where close-to-optimal results are achieved for depth 2. Increasing this to depth 3 brings insignificant performance improvements, but significantly increases the parameter count – e.g. from 4611 to 8771 for width 64. Therefore, we selected a depth of 2 for the experiments in this work.

The hybrid classical-quantum models in Figure 19(b) do not exhibit a comparable saturation behavior for the considered model sizes. With 4 layers, the sample complexity reduces for qubit counts from 6 up to 14. Moreover, the model size in term of parameters only grows slowly for these instances, i.e. from 189 to 437. However, at this point we are faced with a current technical bottleneck: While current quantum hardware is not robust enough to run the quantum models with high enough fidelity, classical simulation costs increase exponentially with qubit count. The simulation time increased additionally by a large factor, as we have to execute 100 full training runs for a single data point. In Table 3 we summarize runtimes that can typically be expected for training the different models in simulation. While the results suggest that it is possible to reduce the sample complexity even further by increasing the qubit count, experimental validation is currently infeasible. However, already these comparatively small quantum models are competitive with much larger classical models. This highlights the promising potential of scaling the quantum approaches, once the hardware development has caught up. Increasing the number of layers seems to have a less significant impact on the overall performance, but still some gains might be possible there. For the experiments in this work, we selected a moderate size of 4 layers.

Overall, we conclude that the classical baseline with width 64 and depth 2 is the sweet-spot model size w.r.t. sample efficiency. For the quantum model the best performance was achieved with 14 qubits and 4 layers, but it is reasonable to assume that further improvements are possible. For saving computational resources, for most experiments we reduced the qubit count to 10. Therefore, the results in this work could be interpreted as comparing an close-to-optimal classical model to an only partially optimized quantum model with potential for future improvement.

## E. Task Complexity of `BeamManagement6G` Environments

In this appendix, we discuss two different ways to adjust the difficulty of the developed `BeamManagement6G` environment. So far, we conducted our experiments on the environment configuration in Figure 11, which contains 3 antennas. Furthermore, the trajectories were sampled using 3 support points, we also refer to this setup as trajectories of *degree* 3. In the following, we will vary both these setup parameters and observe the change in task complexity. We interpret the averaged sample complexity, exhibited across various models, as task complexity. While this might not quantitatively capture the ground-truth difficulty across all possible solution methods, it suffices for a qualitative comparison. The results are summarized in Figure 20.

First, we use the standard environment configuration but increase the trajectory degree up to an value of 6. Intuitively, this leads to more complicated and unpredictable trajectories, which should complicate the task for the RL agent. Examples of

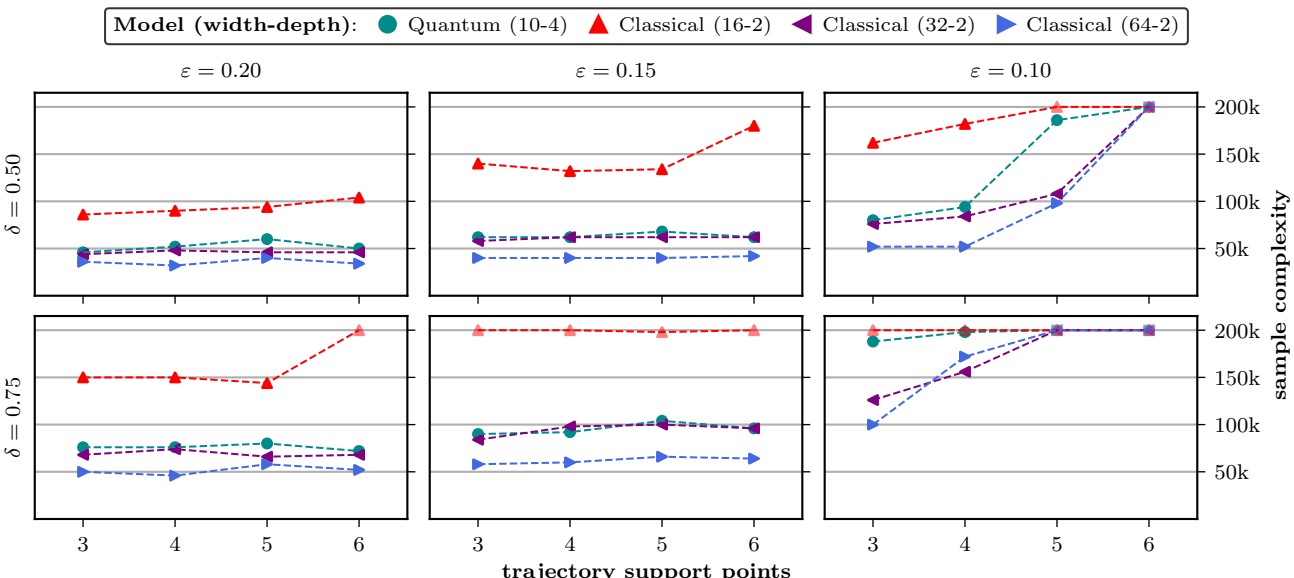

(a) Sample complexity of the standard `BeamManagement6G` environment with increasing trajectory degree, examples see Figure 14.

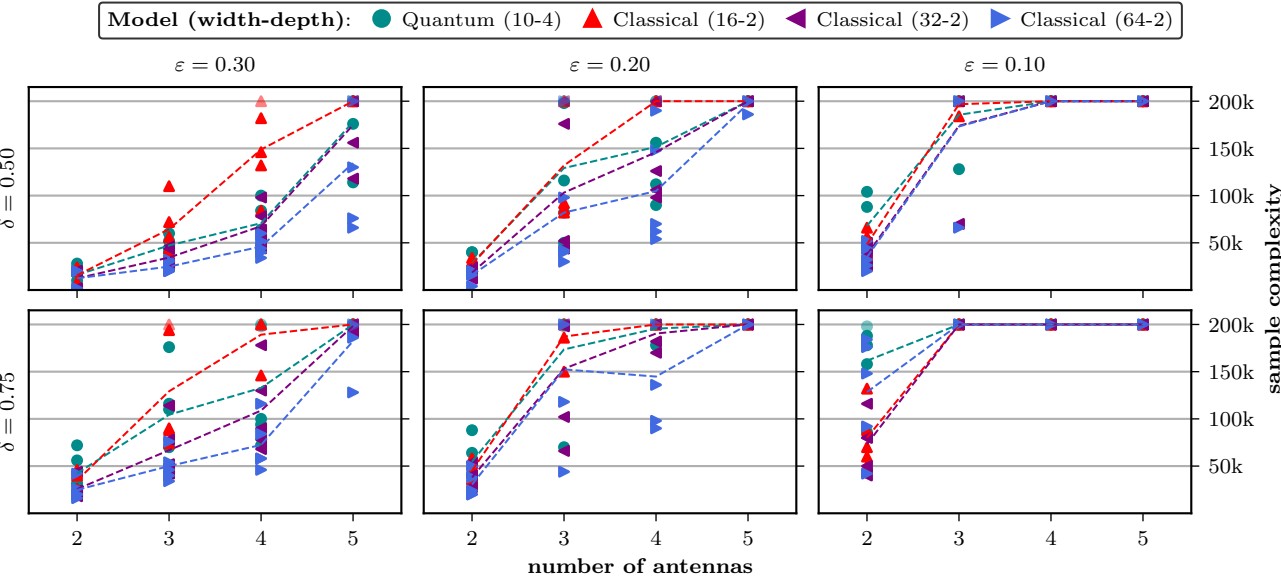

(b) Sample complexity of random `BeamManagement6G` environments with increasing number of antennas. The dashed lines depict the average over 5 environments for each antenna count, for plots of the used instances see Figure 13.

*Figure 20.* Task complexity of different variants of the `BeamManagement6G` environment in terms of the sample complexity exhibited by various classical and a hybrid DDQN algorithms: (a) depicts this scaling behavior for instances of the standard 3-antenna environment from Figure 11; (b) shows the sample complexity over 5 instances with 2, 3, 4, and 5 antennas each, with a trajectory degree of 3; The model width denotes the number of neurons in classical hidden layers, and the number of qubits in quantum models, respectively. Moreover, model depth refers to the number of hidden layers in DNNs, and number of ansatz repetitions in VQCs. The values are averaged over these instances to account for the randomness in antenna placement. All data points are calculated from 100 runs as before.

trajectories with varying number of support points can be found in Figure 14. As expected, in Figure 20(a) we can observe an increase of sample complexity over all model instances. While this behavior is less visible for larger threshold probabilities $\delta$ and larger error thresholds $\varepsilon$, for decreasing values of both the behavior get significant. Moreover, this replicates the behavior from the rest of this work, that the quantum model outperforms the similar-sized classical model, i.e. width 16 and depth 2, on all instances. Furthermore, for most $\varepsilon$-$\delta$-configurations the performance closely matches that of the larger classical models. Note, that this is only the 10-qubit quantum model, i.e. one can expect performance improvement for 14 qubits, especially for the degree 5 setup. However, due to the large overhead of generating the raw results for this setting, such considerations are too computationally expensive. For a trajectory degree of 6, the movement of the UEs becomes too unpredictable for all models to allow for informed antenna selection. Looking at samples of such trajectories in Figure 14 also suggests that a degree of at most 5 should be used to model human behavior. Overall, it is reasonable to claim that with increasing trajectory degree also the underlying task gets more difficult.

Second, we modify the actual placement and orientation of the antennas in Figure 20(b). In all instances the trajectory degree is set to 3. We sample random instances for 2, 3, 4, and 5 antenna positions from $[0.0, 6.0] \times [0.0, 6.0]$ as explained in Appendix A.2, i.e. by enforcing an Euclidean distance of at least 1.5 between pairs of antennas. The direction is assigned uniformly at random. Some of the resulting configurations can be found in Figure 13, all 5 for each antenna count are available in the GitHub repository. As the environments seem to become significantly more difficult with increasing antenna count (compared to increasing the trajectory degree) we relaxed the error thresholds $\varepsilon$. As there is a lot of randomness involved for the actual environment setup, we averaged the sample complexity over all 5 instances, in order to get a more robust estimate. For all models, one can see a clear increase in sample complexity with increasing antenna number. Similar to above, the quantum model exhibits a competitive performance for most $\varepsilon$-$\delta$-configurations. Only for $\delta = 0.75$ and $\varepsilon = 0.10$ the performance on the 2-antenna environments is inferior to the similar-sized classical model. However, for this setup also the usually best-performing classical model with a width of 64 seems to struggle. Moreover, also for this setting we could only simulate sufficient runs for the 10-qubit model. For 5 antennas, all but the large classical model on one environment instance fail to reach the desired quality threshold. This is not too surprising, given the complex interference patterns in Figure 13. To improve upon this, one might need to resort to larger models, or provide additional information to the agent. This can e.g. be done by stacking multiple of the past RL states. However, as the reported results are sufficient to see a clear trend w.r.t. task complexity, this is out of the scope of this work. To summarize, there is a clear correlation between the number of antennas placed in the environment and the resulting task complexity.

As discussed, both the trajectory degree, and the number of antennas can be used to adjust the difficulty of the `BeamManagement6G` environment. Furthermore, the clear real-world inspiration and the sound physical dynamics enhance the practical relevance. Overall, we are confident that the `BeamManagement6G` environment lends itself as a sophisticated benchmark for quantum reinforcement learning (QRL) with the possibility to create instances of increasing complexity.

