# OpenReview forum: "Benchmarking Quantum Reinforcement Learning"
_ICML.cc/2025/Conference — ICML 2025 poster_

### Official Review · Reviewer_nWH6 · 2025-03-12

**Overall Recommendation:** 2

**Summary:**

This paper addresses the issue of valid performance comparisons between classic RL and QRL.  The authors propose a benchmarking methodology, which is based on a statistical estimator for sample complexity and a definition of statistical outperformance. In addition, a novel RL benchmark is established on which they compare the performance of DDQN and PPO using classical neural networks and VQCs. The results of the experiments show that the quantum variants consistently outperforms the classical version when the number of trainable parameters is similar.

**Claims And Evidence:**

The claims seem to be supported for the setup of the experiments.

**Essential References Not Discussed:**

There are two recent publications in the area of QRL that could be added to the literature review:

QRL algorithm with provable advantage:
Wiedemann, Simon, et al. "Quantum Policy Iteration via Amplitude Estimation and Grover Search–Towards Quantum Advantage for Reinforcement Learning." Transactions on Machine Learning Research. 2023.

Model-based Offline QRL:
Eisenmann, Simon, et al. "Model-based Offline Quantum Reinforcement Learning." 2024 IEEE International Conference on Quantum Computing and Engineering (QCE). Vol. 1. IEEE, 2024.

**Experimental Designs Or Analyses:**

The hyperparameters used in the experimental design for the RL algorithms are good choices. As mentioned above, I’m not convinced that this benchmark is a good choice for their experiments.

**Methods And Evaluation Criteria:**

I don’t understand why the only benchmark in this paper is a completely new benchmark. Why not using at least one or two well established RL benchmarks like inverted pendulum or cart-pole?
I’m not convinced that the new BeamManagement6G benchmark is particularly useful RL benchmark.
1.	Is it even an RL problem? As I understand, the agent can select any of the antennas at any time, and there is no planning ahead of the very next step needed. Such problems are called contextual bandits and can be seen as RL tasks with gamma=0. The agent’s only task is to select the best next antenna given the current context (state).  Can you explain why discount values of 0.9, 0.95, or 0.99 are needed?
2.	The true Markov state is only partially observable. The agent is limited to use only the current values of Antenna, Codebook, and Intensity. But why? If the agent would be allowed to look further into the past it could estimate the direction and velocity of the mobile phone way better.

**Other Comments Or Suggestions:**

Sometimes “Fig. X” is used and sometimes “Figure X”
Figure 1 is placed on the front page but mentioned for the first time on page 6.

**Other Strengths And Weaknesses:**

N/A

**Questions For Authors:**

N/A

**Relation To Broader Scientific Literature:**

I think the premise of this paper is very interesting and important: find a good common basis to compare RL and QRL methods to yield solid results from which statements about advantages and disadvantages can be derived.

**Theoretical Claims:**

No.

---

> ### Author Rebuttal · Authors · 2025-03-28
>
> We thank the Reviewer for the assessment of our paper. We appreciate the constructive comments of the reviewer and are confident that addressing them has further strengthened the quality of our work.
>
> - Regarding **Methods And Evaluation Criteria**:
> 	1. An argument for why the BeamManagement6G environment is particularly meaningful for QRL models is outlined in the rebuttal for Reviewer **jcrL**. To show the validity of our methodology for more diverse environments, we repeat the valuation for the CartPole environment in the revised version of the manuscript, as discussed in the rebuttal for Reviewer **Vo3S**
> 	2. On the formulation of the beam management task as MDP / RL environment please see the argument at the bottom.
> 	3. As the Reviewer correctly points out, including several previous observations in the policy input can provide some (implicit) indication on the UEs direction and speed of movement, thus making action selection easier. We emphasize that treating the information from several previous time steps as observation is exactly what we do. This allows the agent to infer the current direction of its motion. We will clarify this detail in the updated version of the manuscript.
>
> - Regarding **Essential References Not Discussed**: Thanks for drawing our attention to these references, we already incorporated them into Section 2 for a updated version of our paper. One comment: While from our perception the first one is targeted for fault-tolerant quantum computing, i.e. a somewhat different setup than we consider, the guarantee of advantages related to sample complexity is quite interesting.
>
> - Regarding **Other Comments Or Suggestions**: We adapted the use of referencing to be consistent. We acknowledge that the first referencing of Figure 1 being delayed to page 6 is not optimal and will adapt that. However, we would opt for keeping it on the front page, as we think the plot conveys the core concept regarding our sample complexity estimator.
>
> In general, we want to thank the Reviewer again for the detailed suggestions, and are especially happy of the assessment ``I think the premise of this paper is very interesting and important: find a good common basis to compare RL and QRL methods to yield solid results from which statements about advantages and disadvantages can be derived.`` We think that this perfectly summarizes the intention behind our work and hope this or a similar rigorous benchmarking procedure will be established in the realm of quantum reinforcement learning. Finally, we hope that the changes and extensions we discuss above improve the quality of our work from your point of view.
>
> ---
>
> ## On the Formulation of the Beam Management Task as an MDP/RL Environment:
>
> We acknowledge that we did not fully clarify the subtleties of the BeamManagement6G environment and why it constitutes a valid RL benchmark. Our environment models handover management problem (selecting the most suitable base station) not the beam selection problem per base station. As described in Appendix A, the trained policy selects the optimal base station, while a low-level algorithm (outside the policy) selects the best beam from the available codebook at that base station.
>
> At each time step, the agent observes the selected base station, the beam ID, and the beam intensity received. The choice of the base station (the chosen action) results in different next states (different base stations lead to different next beam ID and received intensity). Therefore, this setting is not a contextual bandit problem since the state reached after taking an action is dependent on the action taken and by that determines the future state-action sequence.
>
> Whether to RL or not to RL (https://arxiv.org/abs/2405.19045, https://arxiv.org/abs/2401.14823) has been discussed extensively for the problem of handover management:
> Although the policy lacks an explicit planning component (without knowledge of the UE's future movements or the low-level beam selection algorithm at each base station), it must consider the future effects of its actions (i.e., the discount factor $\gamma$ cannot be zero) for several reasons:
> - Ping-Pong Effects: Incorrect base station selection can lead to lower beam intensities and may require reconnection to the previous base station. Such ping-pong effects hinder network and UE efficiency.
> -  Timing of Handovers: The policy must implicitly determine the optimal timing to connect to a new base station and disconnect from the current one to maintain high beam intensity values, ensuring consistent quality of service during handovers.
> -  Intermediate Base Stations: Although not penalized in this work, connecting to an intermediate base station between two others may provide minor benefits for a few time steps.
>
> We also think that the exact definition of the task should not be too important for the main contributions of our paper: We establish the robust benchmarking procedure in an environment-independent way.

---

> > ### Comment · Reviewer_nWH6 · 2025-04-09
> >
> > Thank you for your response. I believe that without a penalty for actions in the benchmark, the optimal policy should select the antenna that maximizes the immediate reward. It seems that the paper might primarily serve as a means to promote this new benchmark. Additionally, I am unable to review the promised experiments and results related to the cart-pole task.

---

> > > ### Author Response · Authors · 2025-04-09
> > >
> > > We thank the reviewer for their further comments and would like to highlight two additional points:
> > >
> > > - We want to stress that the main emphasis of the paper is not the environment we use, but the benchmarking methodology including the statistical estimator for sample complexity. The BeamManagement6G environment serves a replaceable example environment (with particular merits for benchmarking QRL as discussed in the Rebuttal for Reviewer **jcrL**).
> > > - The results of the CartPole experiments have been uploaded to https://zenodo.org/records/15097065?token=eyJhbGciOiJIUzUxMiJ9.eyJpZCI6ImU0MzE3YTcwLTY3OTMtNDRmOC1iYjJiLWQ0N2EwNTExNTBjNyIsImRhdGEiOnt9LCJyYW5kb20iOiIwNjliOTVjZWM4MTM5NTA1ZTQ4NzhkMjJmZGVlMzU5YSJ9.iIKBA2-L10uYnzHsE4ibgLj3A4OaD73cVatmN-Unu_r78aOP3w8ZEArx1qvjW9C502Wlvo-10E5dYNkfMG52sQ. Further details on the experiments are also outlined in the Rebuttal for Reviewer **Vo3S**.

---

### Official Review · Reviewer_Vo3S · 2025-03-13

**Overall Recommendation:** 4

**Summary:**

The paper introduces a standard benchmarking methodology for quantum reinforcement learning (QRL) algorithms, executed with high statistical rigor. The proposed methodology emphasizes sample complexity as a key metric and introduces a statistical estimator for empirical sample complexity. Additionally, the authors define a notion of statistical outperformance to facilitate fair comparisons between algorithms. The benchmarking framework is applied to the BeamManagement6G environment, and the results are well-visualized, providing clear insights into the comparative performance of different QRL methods. The authors also provide code, ensuring reproducibility.

### Update after rebuttal:
The authors have addressed my main concerns effectively. Their clarification on the estimator’s properties, expansion on related RL benchmarking literature, and initial results on an additional environment (Cartpole) strengthen the paper. The open-source implementation and careful framing of quantum advantage claims further support their contributions.

**Claims And Evidence:**

The authors claim that their benchmarking methodology enhances reproducibility and standardization in the fragmented field of QRL. This claim is well-supported by the detailed description of their statistical approach, including the empirical sample complexity estimator and statistical outperformance metric. The inclusion of publicly available code further strengthens the credibility of their claims by allowing independent verification.

**Essential References Not Discussed:**

The paper appears to cite relevant prior work on sample complexity and benchmarking in QRL.

**Experimental Designs Or Analyses:**

The experimental setup is well-structured, with clear definitions of benchmarking criteria and performance metrics. The use of BeamManagement6G as a test environment is justified, and the results are well-visualized and easy to interpret. However, further validation on additional QRL environments would enhance the generalizability of the proposed methodology.

**Methods And Evaluation Criteria:**

The paper employs sound statistical analysis and machine learning techniques to develop a robust benchmarking methodology. The choice of sample complexity as a core metric aligns well with existing evaluation methods in reinforcement learning. The proposed statistical estimator and outperformance metric offer a meaningful and transparent way to compare QRL algorithms. The evaluation is well-documented, and the methodology is appropriate for the problem domain.

**Other Comments Or Suggestions:**

Clarify whether the benchmarking methodology can be easily adapted to other QRL environments beyond BeamManagement6G.

**Other Strengths And Weaknesses:**

Strengths:
- The introduction of a standardized benchmarking methodology is a significant contribution to QRL research.
- The statistical rigor in defining sample complexity and outperformance metrics is commendable.
- The clear presentation and well-visualized results make the findings accessible and interpretable.
- The inclusion of code ensures reproducibility.

Weaknesses:
- The paper primarily focuses on BeamManagement6G; validation on additional environments would enhance its impact.
- While the empirical approach is strong, further theoretical justification of the proposed statistical estimator would be beneficial.
- A broader discussion comparing QRL benchmarking with classical reinforcement learning benchmarks could improve contextualization.

**Questions For Authors:**

How does the proposed benchmarking methodology compare to classical reinforcement learning benchmarking approaches in terms of adaptability and standardization?

Do you plan to validate this methodology on additional QRL environments, and if so, what are the main challenges in doing so?

**Relation To Broader Scientific Literature:**

The paper builds upon prior work on sample complexity in QRL but offers a novel benchmarking approach with enhanced statistical rigor. While it acknowledges previous efforts in QRL evaluation, a more detailed discussion on how this methodology compares with other benchmarking efforts in classical reinforcement learning would be valuable.

**Theoretical Claims:**

The paper does not focus heavily on theoretical proofs but rather on empirical validation. The statistical framework introduced for sample complexity estimation appears well-founded. However, a more formal proof or derivation of the estimator’s properties could strengthen the theoretical contributions.

---

> ### Author Rebuttal · Authors · 2025-03-28
>
> We thank the Reviewer for the positive assessment of our paper. We are sure that we can address the comments sufficiently in a camera-ready version of the paper:
>
> - Regarding **Theoretical Claims**: You advocated for a more formal proof of the estimator's properties. We have derived key properties like consistency and bias in appendix B, and also empirically analyzed the approximation quality wrt the central limit theorem.
>
> - Regarding **Relation To Broader Scientific Literature**: We are working on extending our discussion on related benchmarking efforts of classical RL in Section 2 of our paper. Among others, we now cover works like e.g. https://proceedings.mlr.press/v119/jordan20a.html.
>
> Furthermore, we want to address the stated questions:
>
> - Currently, there is unfortunately no standardization in QRL. Our work may establish a standard for measuring sample complexity, but more work is needed for other metrics.
> - Indeed the adaption to other QRL/RL environments is straightforward, as our implementation (available and open-source) makes use of the gymnasium environment structure; as discussed below, we are currently in the process of generating results for the readily available Cartpole environment, for details see below.
>
> In general, we want to thank the Reviewer again for the detailed suggestions. We hope that the changes and extensions we discuss above improve the quality of our work from your point of view.
>
> ---
>
> ## On the extension to other environments:
>
> In our initial research we performed experiments on the standard Cartpole environment with  vanilla policy gradient. We observed some superiority of quantum models over classical ones, consistent with claims in the QRL community. However, we emphasize that we do not claim this as quantum advantage, as elaborated below:
> - We appreciate the reviewers' suggestion that including results on more well-known environments would strengthen our paper's clarity and generalization. We have uploaded two figures addressing the Cartpole environment to  https://zenodo.org/records/15097065?token=eyJhbGciOiJIUzUxMiJ9.eyJpZCI6ImU0MzE3YTcwLTY3OTMtNDRmOC1iYjJiLWQ0N2EwNTExNTBjNyIsImRhdGEiOnt9LCJyYW5kb20iOiIwNjliOTVjZWM4MTM5NTA1ZTQ4NzhkMjJmZGVlMzU5YSJ9.iIKBA2-L10uYnzHsE4ibgLj3A4OaD73cVatmN-Unu_r78aOP3w8ZEArx1qvjW9C502Wlvo-10E5dYNkfMG52sQ (anonymized) and will incorporate them into the camera-ready version. To explain:
> 	- **Figure 1** compares classical and quantum models for the CartPole-v1 environment. Interestingly, the quantum model achieves lower sample complexity across all epsilon-delta configurations. We performed extensive hyperparameter tuning and tested classical models ranging from 30 to ~17K parameters, reporting the best one we identified. While this indicates some superiority, we explain below why this shouldn't be considered quantum advantage.
> 	- **Figure 2** provides a cross-section of Figure 1 at a reward of 475—the threshold typically considered as solving the Cartpolev1 environment. It illustrates that the best model depends on the desired success probability and that results may not always be significant.
> - We want to highlight that, although these plots suggest some superiority of quantum methods, it's crucial to phrase conclusions carefully. We attribute the observed behavior to the inductive bias of quantum models for this problem. In contrast, many current QRL works claim quantum advantage without sufficient statistics and with around 10 runs (the above plots average over 100). Our paper advocates for more rigorous testing of such claims and provides tools to facilitate this.
> - In conclusion, we believe the Cartpole environment (and most standard RL benchmarks) is insufficient for rigorous testing of QRL. The performance depicted above was achieved with a quantum circuits on only 4 qubits, which are easy to simulate. Moreover, the environment isn't adaptable for meaningful scaling analysis, unlike the BeamManagement6G environment we studied. As summarized in Sections 7 and 8, we cannot definitively answer whether QRL provides actual quantum advantage—this requires further analysis on larger systems. However, we are confident that our work offers valuable tools for such benchmarks, once more efficient quantum circuit simulation methods are available or when execution on quantum hardware becomes possible.

---

### Official Review · Reviewer_3L1D · 2025-03-13

**Overall Recommendation:** 3

**Summary:**

This paper analyzes quantum reinforcement learning to provide a more nuanced evaluation on the potential for quantum advantage. They introduce a sample complexity metric that aims to tackle the evaluation issues, and perform a number of empirical simulations on a new RL environment to evaluate the potential for quantum RL in different regimes.

**Claims And Evidence:**

The core claims of this paper (regarding the nuance required when analyzing quantum RL) are generally clear and convincing. Potential improvements will be outlined below.

**Essential References Not Discussed:**

The analysis of RL algorithms is the subject of quite some interest, and it would benefit the authors to contextualize their proposed evaluation more against this backdrop. Additionally, some literature like https://arxiv.org/abs/2006.16958 is missing (especially given this CDF is quite reminiscent of their sample efficiency.

**Experimental Designs Or Analyses:**

In general, the experiments are sufficient.

**Methods And Evaluation Criteria:**

I’m not sure I’m sold on the new environment. I agree it is valuable to have an environment that is small that you can also tune the complexity and scale of readily (although I would say many of the minigrid/gridworld environments fit that niche). The industrial relevance of the environment doesn’t matter to this paper as well. The real issue is that the environment decreases reader comprehension of the main points of the paper. There is no intuition on how hard or meaningful results are in this environment, so it can be hard to really build an understanding of the impact of the comparisons. If the authors are attached to this environment, then at the very least, they should include other environments that the paper they cite use (e.g. the usual CartPole, BlackJack, minigrid, mountain car etc.). This would both help reader understanding by providing the analysis in context of something they already know, and also help verify the generality of the results (if things are only checked on one environment, maybe it’s just a phenomenon of that environment and doesn’t apply to others?).

**Other Comments Or Suggestions:**

From a purely constructive criticism perspective (this has no impact on my evaluation of the paper), I’m not sold on Figure 1. I get what the authors are going for, but it takes up a lot of space and doesn’t actually say much beyond (small C < small Q < big C). I don’t really have an idea for a better figure, but I believe there is one that could have higher information density.

I also feel the same about figure 6. It seems like this should just be a table, the visual information of this is conveying only a small collection of discrete points (and the colors/labels don’t offer anything beyond what text would in a table).

**Other Strengths And Weaknesses:**

This paper does a lot of things right, and I think it serves an important role in the QRL literature. However, I think there are some things that are holding it back, but could be readily improved upon. Some of which has already been discussed.

I don’t think figure 8 adds much. It just says “classical and quantum both solved it and random doesn’t”, but that was already conveyed in the text (and is generally true of RL environments). I think there is a place here for a connection to a more traditional RL figure. I know there are issues, as pointed out by this paper, on the “iteration vs reward” plots, but it could be good to connect what people know from RL literature to this new framework (for example, if I see sample complexity of [X], how would that reflect itself in reward graphs). This figure could also just be cut entirely.


This is not necessarily unique to this paper, but the focus on quantum parameter count could use more justification. What I mean is that, classical parameter count only matters because it is a semi-useful proxy for speed (or really energy, since there might be cases where we can use more processors in parallel and gain more speed at the cost of energy) which is why there’s been a big shift away from parameter counts towards things like FLOPS (also with the advent of modern ML methods, the correlation between param count and speed was beginning to loosen). Now perhaps the same is somewhat true for quantum, more layers does take more time, and more qubits do take more energy to control, but I’m not sure the comparison between parameter counts means much. This is not an argument with model complexity, I agree the plots regarding parameters vs. model complexity are fair. But the general comparison of “small quantum” vs. “small/large classical”. The number of parameters don’t really matter, it’s only the advantage that matters (either via time, or energy, or sample complexity), so the idea of comparing 400 quantum vs 400 classical parameters doesn’t really seem meaningful (because those parameters are doing different things and we are operating in very different regimes of computation). Comparing quantum vs. classical in general is important, but the focus of papers on saying “small Q is better than small C” as if that’s meaningful (like we don’t care about the number of parameters, just about how it performs on those three axes) should be more addressed, specifically focusing the parameter discussion around model complexity and comparisons around sample/time.

**Questions For Authors:**

1. Presumably in figure 7 quantum will saturate too? Also why not do depth rather than width? I see that analysis in Figure 18, but it looks flat, which seems to indicate to my point that quantum parameters aren’t interpretable in the same way.

**Relation To Broader Scientific Literature:**

This paper builds upon the broad literature of QRL which has recently been focused on hybrid approximations using traditional RL algorithms. It suggests a number of improvements in the analysis of this work, but could be more forceful in its suggestions and analysis (as much of the existing literature is a number of problems well beyond what many RL papers have).

**Theoretical Claims:**

The focus was on empirical work, any theoretical claims are readily checked.

---

> ### Author Rebuttal · Authors · 2025-03-28
>
> We thank the Reviewer for the assessment of our paper and are pleased to read that the reviewer thinks that `This paper does a lot of things right, and I think it serves an important role in the QRL literature.` We appreciate the constructive comments and are confident that addressing them has further strengthened our paper.
>
>  - Regarding **Methods And Evaluation Criteria**: We agree with the reviewer that it can be difficult for the reader to get intuition on e.g. the hardness of the BeamManagement6G environment. Nevertheless, we believe that the environment is particularly meaningful for benchmarking RL vs. QRL as we argue in the rebuttal for Reviewer **jcrL**. Following the reviewer's suggestion we included the application of our evaluation methodology to the CartPole environment, see the rebuttal for Reviewer **Vo3S**.
>
> - Regarding **Relation To Broader Scientific Literature**:
> In the revised version of the manuscript we added a paragraph were we clearly state suggestions on how to achieve robust and reproducible benchmarking results based on the methodology we introduced and stress its importance for standardization and to avoide false claims.
>
> - Regarding **Essential References Not Discussed**: We have extended our discussion in Section 2 with the suggested paper and are incorporating additional references on benchmarking classical RL (e.g. arXiv:2402.03046, arXiv:2209.12016)
>
> - Regarding **Other Weaknesses**:
> 	1. We agree, that Figure 8 does not add much value. We think it makes sense to move it to the appendix--depending on space--and instead incorporate the above mentioned results on Cartpole into the main part.
> 	2. We agree with the reviewer that focusing on parameter count isn't always adequate; e.g. the difference between models with 387 quantum and 4,611 classical parameters is less relevant due to simulation overhead (since training the quantum model takes more time than the classical model).
>     Indeed the parameter count by itself is of limited value, only metrics like e.g. energy, time and sample complexity matter. The key point we want to convey is that parameter count can be used to define a sequence of models in model space (assuming additional hyperparamter search for given parameter count). This sequence can then be used to identify potential trends for example on the sample complexity axis as we scale to more powerful quantum models. We made this point more precise in the modified version of the manuscript. Comparing parameter counts can still have merit also for ther axis like time and energy, because:
> 	   - Using parameter count as a proxy (e.g., in Figure 7), we speculate that scaling quantum models may provide a net gain in sample complexity. Validating this requires larger-scale empirical analysis beyond our computational resources. This limitation explains why most QRL work focuses on fewer than 10 qubits, and our paper questions advantage claims in this regime.
> 	   - For complex problems, large classical models become expensive and time-consuming to train. If similar performance would be achievable with significantly fewer quantum parameters, executing quantum circuits could potentially be more efficient in terms of time and cost as quantum hardware matures.
>
> - Regarding **Other Comments Or Suggestions**: We agree that the information content of Figure 1 and 6 is limited, especially as both describe the same setup. With the new empirical results on the Cartpole environment, we can replace one of the figures with a new one providing more intuition.
>
> - Regarding the **Questions**:
>     1. We acknowledge this valid concern. Saturation is expected but might occur at lower sample complexity for quantum models than for classical models. Though speculative, the performance gap to the best classical model at higher qubit counts is small.
>     The 14-qubit model pushed our simulation capabilities to the limit due to the 100 runs needed per model for robust analysis; simulating the circuit is feasible, but gradient computation and training are very expensive. Nearly a quarter of the 40,000 compute hours were devoted to these systems. However, our work provides benchmarking techniques, once more efficient simulations or suitable quantum hardware is available.
>
>     2. We focus on scaling qubit number since performance saturates quickly with increased depth, as you pointed out. While quantum parameters certainly require different interpretation than classical ones, we do not think that this can be stated based on Figure 18. Indeed, also with scaling of classical depth (hidden layers) the performance saturates quickly, i.e. we see the same behaviour as for the quantum models.
>
> We thank the Reviewer again for the detailed suggestions. We think that currently there are a lot of issues in the QRL literature, of which we want to address at least a small fraction. We hope the discussed revisions and extensions improve the relevance and quality of our work from your perspective.

---

> > ### Comment · Reviewer_3L1D · 2025-04-02
> >
> > I appreciate the authors response. I think they have addressed many of my issues, and the changes they will make will improve the paper. The results from Vo3S response (on CartPole) are very influential for these improvements. I understand the motivation for the new environment, although I still remain hesitant on its necessity (although new environments often are the subject of their own papers, e.g. https://arxiv.org/abs/2402.16801 I suppose this environment is too small). In general, it seems like a confounding variable in otherwise important results. But I understand that a small environment with flexible scaling might be a gap that is needed, so as long as the environment is well maintained and integrated and easy to use in existing QRL gym workflows, I suppose it can serve a role. Looking at other reviewers, this seems like a common thread, so hopefully there can be some collective agreement and resolutions to be found.
> >
> > I have updated my score to reflect these improvements.

---

> > > ### Author Response · Authors · 2025-04-07
> > >
> > > We thank the reviewer for acknowledging our rebuttal and their suggestions in the original review. We agree that the added Cartpole results significantly strengthen our paper.
> > >
> > > Regarding the points on the environment:
> > > - Encoding into quantum states introduces specific requirements satisfied by few existing environments; for example, the environment in arXiv:2402.16801 would be challenging due to its image representation.
> > > - We have detailed the availability of our BeamManagement6G environment in the data availability statement of the camera-ready version. It strictly follows the Gymnasium API and should be straightforward to use in other work.

---

### Official Review · Reviewer_jcrL · 2025-03-19

**Overall Recommendation:** 3

**Summary:**

The paper presents a benchmarking methodology for quantum reinforcement learning (QRL) by introducing a statistical estimator for sample complexity and a robust definition of statistical outperformance. Through experiments in a novel benchmarking environment inspired by wireless communication tasks, the study finds that hybrid quantum-classical models can outperform similarly sized classical models but do not definitively surpass larger classical models. The results highlight the need for statistically rigorous evaluations in QRL and question some claims of quantum advantage. The study concludes that while QRL shows promise, its superiority remains uncertain and requires further large-scale empirical investigation.

**Claims And Evidence:**

The paper provides robust statistical evidence—through extensive experiments and resampling techniques—for its benchmarking claims, particularly that hybrid quantum–classical models can achieve lower sample complexity than similarly sized classical models. However, the broader claim of a definitive quantum advantage is less convincing. Specifically, the comparison may not fully account for optimal tuning of classical baselines, and the experiments are limited to small-scale, simulated environments. This makes extrapolating the results to real-world, large-scale problems somewhat speculative.

**Essential References Not Discussed:**

Reviewer not sufficiently familiar with quantum literature to comment.

**Experimental Designs Or Analyses:**

Experiments seem sound, though comparison to a broader range of RL updates should be considered.

**Methods And Evaluation Criteria:**

The proposed methods and evaluation criteria—such as the statistical sample complexity estimator and the BeamManagement6G benchmark—seem well-suited for comparing classical and quantum reinforcement learning, however the reviewer is not familiar with the relevant literature.

**Other Comments Or Suggestions:**

None for now.

**Other Strengths And Weaknesses:**

The paper is commendable for its creative integration of quantum and classical reinforcement learning through a robust, statistically grounded evaluation framework, and its application to a real-world inspired benchmark enhances its relevance and clarity. However, the study's limited experimental scale and potential under-optimization of classical baselines may restrict the generalizability of its conclusions.

**Questions For Authors:**

None for now.

**Relation To Broader Scientific Literature:**

Reviewer is not familiar with quantum literature.

Authors should perform more experiments with diverse RL updates to determine how they vary in the quantum setting. Additionally, broader discussion of baseline tuning is required.

**Theoretical Claims:**

Did not check.

---

> ### Author Rebuttal · Authors · 2025-03-28
>
> We thank the Reviewer for the assessment of our paper. We appreciate the constructive comments made by the reviewer and believe that addressing them has further strengthened our work.
>
> - Regarding **Claims and Evidence**:
> 	1. We agree on the concern, that the claim of *definitive quantum advantage* is not supported by the empirical results of our work. However, this is not something we want to claim, as we see our paper also as critique on the large collection of literature that makes such statements without rigorous statistical analysis. We also explicitly discuss this in Section 7.
> 	2. Regarding the speculative extrapolation to larger, real-world environments: We acknowledge that the statements on scaling with larger quantum models in Section 6.2 are formulated too strongly. We will make clear that the extrapolation is speculative, and further empirical investigation is required to make it more definitive (for which we unfortunately currently lack the compute resources, as training models with over 14 qubits requires upwards of 1 day per run)
> 	3. Regarding tuning of the optimal baseline: Of course we can not guarantee that the classical baseline is optimal wrt all hyperparameters. However, we conducted an extensive hyperparameter search summarized in Appendix C.1 (2160 hyperparameter configurations for DDQN, 1800 for PPO), which should give reasonable backing for our claims.
>
> - Regarding **Experimental Design Or Analysis**: Overall we consider three *standard* RL algorithms, i.e. VPG, DDQN, PPO; while of course there exist many more, our paper does not focus on benchmarking a broad range of algorithms, but rather provides the overall framework; extension to other algorithms and environments should be straightforward also for third-party researchers with the provided open-source code. An extension to the standard Cartpole environment is discussed in the rebuttal for Reviewer **Vo3S**.
>
> Overall, we emphasize that our paper should not be seen as yet another one claiming quantum advantage on some task with some model. We again thank the author for pointing out that currently there are formulations in the paper that suggest otherwise. Rather, our work provides the tools to establish robust statistical backing for such comparisons between classical and quantum models.
>
> ---
>
> ## On the use of the BeamManagement6G environment:
>
> Our motivation for considering only a single (new) benchmark environment is the focus of our work on establishing a benchmarking methodology rather than on solving specific problems.
> - We think that the BeamManagement6G environment has its justification for our work. First of all, environments like Cartpole or Mountain Car are certainly a good benchmark, but are so simple that they can be *solved* by rather small classical and quantum models (small meaning under 100 parameters), which makes scaling analysis difficult. Other more scalable and complex environments, like Frozen Lake and Atari have the problem that the state spaces are too large for meaningful encoding into a quantum circuit. Some thoughts on properties we in general need for a environment suitable for QRL:
>     - The state and action spaces must be sufficiently small for an efficient encoding into a quantum state.
>     - The dynamics must be sufficiently complex for justifying the use of QRL in the first place.
>     - The state elements should be continuous to facilitate the continuous nature of quantum states, which is not the case for e.g. Gridworld environments. For the BeamManagement6G environment, the energy is conitnuous and the codebook element intrinsically corresponds to continuous beam angles.
>
> - Why the BeamManagement6G environment is a particularly suited real-world task with relevance of our sample complexity estimator:  As we discuss in our paper, sample complexity as a metric is well defined only with respect to a given performance threshold. In the BeamManagement6G environment, different UEs may have different service quality requirements, which translates to a different error threshold epsilon. Different policies for different levels of QoS can be trained, which can be naturally translated to different performance threshold levels, as highlighted in Figure 4 in the paper.

---

### Decision · Program_Chairs · 2025-05-01

**Decision:**

Accept (poster)

**Comment:**

This paper was discussed controversially and even after rebuttal and discussion the assessments ranged from Weak Reject to Accept. The claims are essentially seen as sufficiently substantiated. The rebuttal was able to address most of the points of criticism.

The selection of benchmarks continues to be criticized, particularly the use of a previously unknown benchmark.
The criticism remains that the initial submission did not reach this level of quality and that extensive changes were necessary and have now been made, but can no longer be assessed by the reviewers.